# FRET kinase sensor development reveals SnRK2/OST1 activation by ABA but not by MeJA and high CO$_2$ during stomatal closure

Li Zhang[1†‡], Yohei Takahashi[1†*], Po-Kai Hsu[1], Hannes Kollist[2], Ebe Merilo[2], Patrick J Krysan[3], Julian I Schroeder[1*]

[1]Cell and Developmental Biology Section, Division of Biological Sciences, University of California, San Diego, San Diego, United States; [2]Institute of Technology, University of Tartu, Tartu, Estonia; [3]Horticulture Department, University of Wisconsin-Madison, Madison, United States

*For correspondence:
ytakahashi@UCSD.EDU (YT);
jischroeder@ucsd.edu (JIS)

[†]These authors contributed equally to this work

Present address: [‡]Maize Research Institute, Sichuan Agricultural University, Wenjiang, China

Competing interests: The authors declare that no competing interests exist.

## Abstract

Sucrose-non-fermenting-1-related protein kinase-2s (SnRK2s) are critical for plant abiotic stress responses, including abscisic acid (ABA) signaling. Here, we develop a genetically encoded reporter for SnRK2 kinase activity. This sensor, named SNACS, shows an increase in the ratio of yellow to cyan fluorescence emission by OST1/SnRK2.6-mediated phosphorylation of a defined serine residue in SNACS. ABA rapidly increases FRET efficiency in *N. benthamiana* leaf cells and *Arabidopsis* guard cells. Interestingly, protein kinase inhibition decreases FRET efficiency in guard cells, providing direct experimental evidence that basal SnRK2 activity prevails in guard cells. Moreover, in contrast to ABA, the stomatal closing stimuli, elevated CO$_2$ and MeJA, did not increase SNACS FRET ratios. These findings and gas exchange analyses of quintuple/sextuple ABA receptor mutants show that stomatal CO$_2$ signaling requires basal ABA and SnRK2 signaling, but not SnRK2 activation. A recent model that CO$_2$ signaling is mediated by PYL4/PYL5 ABA-receptors could not be supported here in two independent labs. We report a potent approach for real-time live-cell investigations of stress signaling.

## Introduction

Protein phosphorylation of downstream substrates by protein kinases is a central and critical molecular switch for activation of many cell biological processes (*Ardito et al., 2017*; *Stone and Walker, 1995*). However, investigation of protein kinase activities remains a challenge, particularly real-time measurements in living cells. *In-gel* kinase assays are the most common method for measuring protein kinase activities using the (auto-)phosphorylation state of a kinase or a substrate as indicator of the kinase activity (*Manning et al., 2002*). With this method, it is difficult to track dynamic kinase activity in specific cell types or tissues, and time course measurements in living cells and subcellular analyses are not feasible (*Aoki et al., 2012*). To overcome this drawback, a first Förster resonance energy transfer (FRET) biosensor reporting the activity of cAMP-dependent protein kinase A (PKA) was developed by R.Y. Tsien and colleagues (*Zhang et al., 2001*). The design of a FRET-based protein kinase biosensor includes a phosphorylatable substrate protein domain and a phosphorylation recognition domain that together can drive a conformational change between two fluorophores (*Sample et al., 2014*; *Ting et al., 2001*; *Wang et al., 2005*; *Gao and Zhang, 2008*; *Miyawaki and Tsien, 2000*). In the presence of active protein kinase, the substrate domain is phosphorylated, which results in a high affinity to the phosphorylation-recognition domain. This enhanced affinity between the two domains can trigger the two fluorophores to be in closer proximity to one-another

(*Jones et al., 2013*). Such a conformational change can cause higher FRET efficiency (*Lin et al., 2019*; *Jones et al., 2013*). In plants, only one protein kinase FRET biosensor has been reported to date, that senses plant MAP kinase activity using a MAPK substrate domain (MAP kinase phosphatase, MKP1) and a FHA domain (*Zaman et al., 2019*).

Plants have to cope with diverse and complex abiotic and biotic stresses. Abscisic acid (ABA) is a plant hormone, which plays a central role in plant drought, cold and salinity stress tolerance (*Cutler et al., 2010*; *Raghavendra et al., 2010*; *Zhu, 2016*). SnRK2 protein kinases (sucrose non-fermenting-1-related protein kinase 2s) are members of a plant-specific serine/threonine kinase family that play critical roles in ABA signal transduction (*Fujii et al., 2007*; *Mustilli et al., 2002*; *Yoshida et al., 2002*; *Fujii and Zhu, 2009*). ABA activation of SnRK2 protein kinases has been found in several species including *Vicia faba* (*Li et al., 2000*), rice (*Kobayashi et al., 2004*), *Arabidopsis* (*Hrabak et al., 2003*; *Fujii and Zhu, 2009*), *Glycine soja* (*Yang et al., 2012*) and *Triticum polonicum* (*Wang et al., 2015*). In *Arabidopsis*, there are 10 SnRK2 family members (SnRK2.1–2.10) in the genome. Three of these protein kinases, SnRK2.2, SnRK2.3 and SnRK2.6/OST1, are strongly activated by ABA. The *SnRK2.6/OST1* gene is expressed in stomatal guard cells (*Mustilli et al., 2002*). SnRK2.6/OST1 mediates reduction of stomatal apertures and plays a pivotal role in ABA-induced stomatal closure (*Yoshida et al., 2002*; *Mustilli et al., 2002*). In addition, SnRK2.6/OST1 is genetically required for rapid stomatal closure in response to elevation in the $CO_2$ concentration (*Xue et al., 2011*; *Merilo et al., 2013*; *Hsu et al., 2018*). Furthermore, SnRK2.6/OST1 is also required for methyl jasmonate (MeJA)-induced stomatal closure (*Yin et al., 2016*). However, whether MeJA or $CO_2$ activate OST1/SnRK2.6 protein kinases in guard cells and whether $CO_2$ rapidly activates ABA signaling remains controversial.

Studies have shown Methyl jasmonate (MeJA)-induced stomatal closure in several species (*Herde and Pena-Cortes, 1997*; *Gehring, 1997*; *Suhita et al., 2004*; *Akter et al., 2012*; *Förster et al., 2019*) and that MeJA-induced stomatal closing is impaired in *snrk2.6/ost1* mutants (*Yin et al., 2016*; *Munemasa et al., 2019*). Unexpectedly, however, recent *in-gel* kinase assays with isolated guard cell protoplasts detected no MeJA-induced enhancement of OST1 kinase activity (*Munemasa et al., 2019*).

Classical research demonstrated that ABA enhances $CO_2$-induced stomatal closing and vice versa (*Raschke, 1975*; *Raschke et al., 1976*). In addition, plants have previously been found to have higher basal ABA concentrations in guard cells than in leaf mesophyll cells (*Lahr and Raschke, 1988*). *ost1* mutant alleles have been shown to impair $CO_2$-induced stomatal closing in intact leaves and intact plants (*Xue et al., 2011*; *Merilo et al., 2013*), but not as strongly as ABA-induced stomatal closing (*Hsu et al., 2018*). Unexpectedly, however, recent *in-gel* kinase assays show no $CO_2$ activation of SnRK2 kinases in purified guard cells protoplasts (*Hsu et al., 2018*). Moreover, whether ABA-SnRK2 signal transduction is directly activated by $CO_2$ elevation remains controversial (*Chater et al., 2015*; *Merilo et al., 2013*; *Hsu et al., 2018*; *Dittrich et al., 2019*). The requirement to use guard cell extracts for monitoring OST1/SnRK2.6 activity hampers resolution of this question that is key to understanding how the distinct stimuli $CO_2$, ABA and MeJA merge in mediation of stomatal closing.

*In-gel* kinase assay analyses have shown that ABA rapidly activates OST1/SnRK2.6/AAPK protein kinases using purified guard cell protoplasts (*Li et al., 2000*; *Mustilli et al., 2002*; *Hsu et al., 2018*; *Li and Assmann, 1996*; *Mori and Muto, 1997*). ABA-activated SnRK2 protein kinases phosphorylate several downstream substrates including the SLAC1 anion channel and transcription factors (*Takahashi et al., 2013*; *Munemasa et al., 2019*; *Geiger et al., 2009*; *Furihata et al., 2006*; *Lee et al., 2009*). ABA-responsive kinase substrate 1 (AKS1) is a basic helix-loop-helix (bHLH) transcription factor (*Takahashi et al., 2013*). AKS1 was demonstrated to be phosphorylated by SnRK2s in response to ABA in *Arabidopsis* guard cells, as an endogenous phosphorylation substrate (*Takahashi et al., 2017*; *Takahashi et al., 2013*). Moreover, ABA rapidly induces 14-3-3 protein binding to AKS1 in a phosphorylation-dependent manner in guard cells (*Takahashi et al., 2013*). 14-3-3 proteins are conserved proteins in eukaryotes and known for binding to target proteins upon phosphorylation. 14-3-3 proteins recognize two typical 14-3-3 binding motifs, which in AKS1 are the Ser-30 residue and Ser-157 residue (*Takahashi et al., 2013*). The AKS1 (S30A/S157A) mutant protein does not bind to the 14-3-3 protein GF14phi (*Takahashi et al., 2013*).

Here, using AKS1 and an *Arabidopsis* 14-3-3 protein, GF14phi, we describe the development of a genetically-encoded FRET-based biosensor, named SnRK2 activity sensor (SNACS), that reports

SnRK2 protein kinase activity in live plant cells in real-time. We observe real-time SnRK2 kinase activity in response to ABA in living plant cells. ABA-induced FRET shifts are disrupted in *snrk2.2/2.3/ost1* triple mutant guard cells. In addition, we also demonstrate the effects of protein kinase inhibitors on SNACS in planta, showing that SNACS FRET shifts are reversible and providing evidence that basal ABA signaling exists in guard cells. Moreover, using this sensor, we show that, in contrast to ABA, MeJA and $CO_2$ do not cause SNACS FRET ratio increases in guard cells in real-time in vivo analyses. A model that is consistent with the presented findings in guard cells is discussed, in which basal ABA/SnRK2 signaling is required for $CO_2$ and MeJA responses, but without direct activation of SnRK2 kinases. In addition, gas exchange analyses with ABA receptor quintuple and sextuple mutants, performed independently in two of our laboratories (EM, HK and JIS), could not confirm a recent report that stomatal $CO_2$ signaling is mediated by the PYL4 and PYL5 ABA-receptors (*Dittrich et al., 2019*). Imaging using SNACS enables resolution of single cell-type recordings, compared to the need to laboriously isolate and purify >$10^5$ guard cell protoplasts from >100 leaves for each guard cell *in-gel* kinase assay lane. Taken together, a genetically-encoded FRET-based biosensor is developed that reports real-time basal and stimulus-dependent SnRK2 protein kinase activity in single plant cells.

## Results

### Construction of SnRK2 protein kinase activity sensor

In order to attempt production of a genetically encoded biosensor that would report SnRK2 kinase activity in *Arabidopsis*, we used a 48 amino acid domain surrounding the serine-30 residue of the *Arabidopsis* ABA-RESPONSIVE KINASE SUBSTRATE1 transcription factor (AKS1, At1g51140) (*Takahashi et al., 2013*). Furthermore, an AKS1 binding domain was inserted into the construct encoding the *Arabidopsis* full length 14-3-3 protein, GF14phi (267 amino acids). AKS1 has been shown to be phosphorylated by SnRK2s at the serine-30 residue (*Takahashi et al., 2017*). 14-3-3 protein is known to bind to phosphoserine/threonine-containing proteins through amino acid binding motifs, including RxxpSxP (*Sluchanko et al., 2017*). 14-3-3 protein binds to the serine-30 residue of AKS1 only when Ser-30 is phosphorylated (*Takahashi et al., 2013*). The main components of the designed biosensor from N- to C- terminus end are the YPet (YFP variant) fluorescent protein (*Day and Davidson, 2009*), the 14-3-3 phosphopeptide-binding domain, 244 amino acids of an EV linker, 48 amino acids of the *Arabidopsis* AKS1 transcription factor and Turquoise GL cyan fluorescent protein (*Figure 1A*). As a control, we also generated a SNACS$^{S785A}$ mutant isoform, in which the AKS1-Ser-30 residue is substituted with a non-phosphorylatable alanine (Ala) in the AKS1 domain.

A hypothetical simplified working model of the biosensor is outlined in *Figure 1B*: When the sensor is in an unphosphorylated state, the 14-3-3 domain was predicted to have a low affinity for the AKS1 substrate resulting in a low ratio of YPet to Turquoise GL fluorescent emission under the Turquoise GL excitation. However, when the AKS1 Ser-30 in this sensor is phosphorylated, the 14-3-3 domain was hypothesized to bind to the phosphorylated AKS1 region. Based on this working model, we tested this sensor to determine whether it can function as a readout of SnRK2 protein kinase activity.

We performed in vitro phosphorylation assays. The results showed that the OST1/SnRK2.6 protein kinase can phosphorylate this sensor, but not the SNACS-S785A mutant isoform (*Figure 1C*). Furthermore, the unrelated $Ca^{2+}$-dependent protein kinase, CPK6, did not measurably phosphorylate the wild-type SNACS or S785A mutant (*Figure 1C*). To investigate whether this phosphorylation of SNACS might affect the FRET ratio (YPet emission/Turquoise GL emission), in vitro FRET assays were performed using recombinant SNACS and OST1/SnRK2.6 proteins. SNACS protein was incubated in the presence or absence of OST1/SnRK2.6 protein kinase, and then the fluorescence emission profile was determined in response to excitation at 434 nm. The SNACS showed an increase in YPet emission (average data points, 525–531 nm) by an average of 11.8% and a decrease of Turquoise GL emission (average data points, 471–477 nm) by 12.96% in the presence of OST1/SnRK2.6 compared to the SNACS alone upon Turquoise GL excitation (*Figure 1D*). This in vitro SNACS response appears to be nearly saturated under the imposed conditions, as incubation with two times the concentration of OST1/SnRK2.6 protein caused no further clear shift in the fluorescence

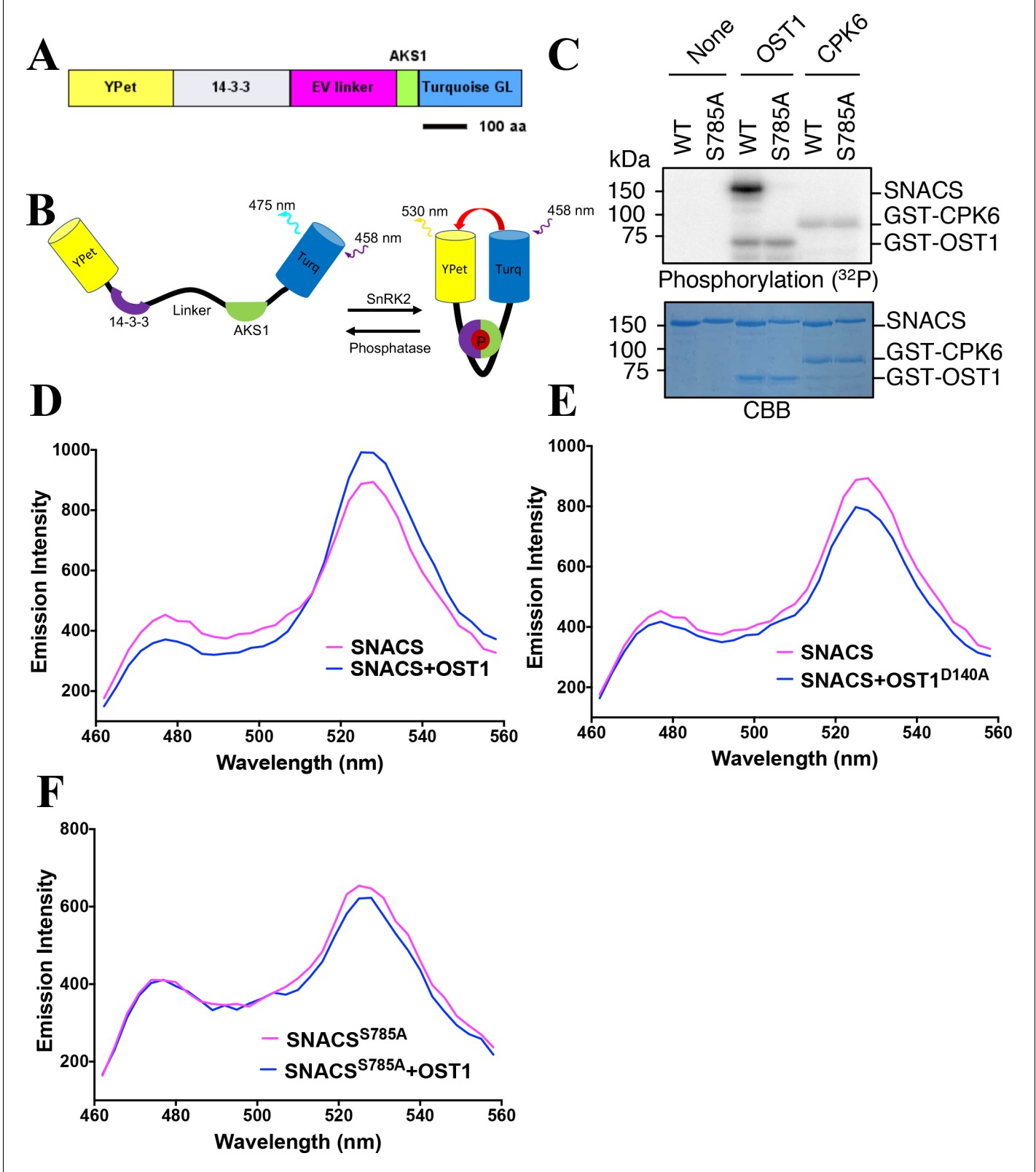

**Figure 1.** Structure and in vitro testing of SnRK2 protein kinase reporter, SNACS, and SnRK2 protein kinase activity. (**A**) Domain structure of the SNACS protein: YPet and Turquoise GL are yellow and cyan fluorescent proteins. The full coding region of the phosphoserine/threonine binding 14-3-3 GF14phi protein (At1g35160) was inserted, EV linker is a 244 amino acid length flexible linker domain, and AKS1 is a 48-amino-acid segment of the

*Figure 1 continued on next page*

Figure 1 continued

*Arabidopsis* AKS1 transcription factor protein. aa, amino acid. (B) Simplified model of SNACS reporter function. Phosphorylation of the sensor within the AKS1 domain is predicted to produce a conformational shift that increases Förster resonance energy transfer (FRET) efficiency due to the enhanced affinity of the 14-3-3 domain for the phosphorylated form of the substrate domain. (C) In vitro phosphorylation assay. SNACS (WT) and SNACS^S785A (S785A) were incubated in the presence or absence of GST-OST1/SnRK2.6 protein kinase or GST-CPK6. Proteins were separated on a gel and the incorporation of $^{32}$P into the substrates was evaluated via autoradiography (upper panel). The lower panel shows a loading control stained with Coomassie brilliant blue. (D) and (E) in vitro FRET assays using SNACS performed in the presence of GST-OST1 or GST-OST1^D140A (inactive OST1 kinase version). SNACS alone controls in D and E were from the same experiments shown in the two panels, as experiments were performed in parallel. The emission spectra of SNACS produced by excitation of the Turquoise GL domain with 434 nm light are shown. (F) In vitro FRET assays using SNACS^S785A mutant reporter performed in the presence or absence of GST-OST1. Emission spectra of the SNACS reporter produced by excitation of the Turquoise GL domain with 434 nm light are shown.

The online version of this article includes the following source data and figure supplement(s) for figure 1:

**Source data 1.** Uncropped gel images for *Figure 1C*.
**Source data 2.** In vitro SNACS FRET ratio values in *Figure 1*.
**Figure supplement 1.** In vitro FRET assay of SNACS reporter.
**Figure supplement 1—source data 1.** In vitro SNACS FRET ratio values in *Figure 1—figure supplement 1*.

---

spectrum (*Figure 1—figure supplement 1E*). In addition, in the presence of the inactive mutant isoform OST1/SnRK2.6 (OST1^D140A) protein kinase as a control, no clear FRET ratio increase was found (*Figure 1E* and *Figure 1—figure supplement 1A,B*). These results suggested phosphorylation of SNACS caused a FRET ratio increase.

Consistent with the above results, the mutant SNACS^S785A isoform with the AKS1 Ser-30-Ala mutation did not show FRET ratio changes after an incubation with the OST1/SnRK2.6 protein kinase upon Turquoise GL excitation (*Figure 1F* and *Figure 1—figure supplement 1C*). In additional experiments, SnRK2.3, another protein of SnRK2 family, induced a SNACS FRET ratio increase as well (*Figure 1—figure supplement 1D*). In contrast, the highly active CPK6 protein kinase (*Brandt et al., 2012*), did not increase the FRET ratio (*Figure 1—figure supplement 1D*). Taken together, these results indicate that SNACS shows an increase of FRET ratio based on a phosphorylation of the AKS1 domain by SnRK2 protein kinases in vitro. Furthermore, the AKS1 Ser-30-Ala mutation in SNACS disrupts this response in vitro (*Figure 1C,F* and *Figure 1—figure supplement 1C*).

## SNACS reports SnRK2 activity dynamics in plant cells

We next investigated the functionality of SNACS in planta by performing live cell imaging. Transient expression experiments using the SNACS reporter in *Nicotiana benthamiana* were performed. SNACS driven by the cauliflower mosaic virus 35S promoter was co-expressed with *pUBQ10:OST1-6xHis-3xFLAG(HF)* (*Waadt et al., 2015*) in *Nicotiana benthamiana* by co-infiltration. Emission ratio images were recorded 3 days after infiltration. We immobilized abaxial intact leaf epidermal tissues on glass-bottom dishes, and then identified epidermal cells showing fluorescence. Interestingly, the application of ABA caused time-resolved emission ratio (YPet to Turquoise GL emission) increases in *N. benthamiana* epidermal cells (*Figure 2A,C,D*) (n = 3 experiments; 11–13 cells imaged in each experiment). As ABA was dissolved in 0.02% EtOH, EtOH was used as a solvent control. EtOH treatments did not induce measurable emission ratio increases of SNACS (*Figure 2B*, n = 3 experiments; 4–7 cells imaged in each experiment).

SNACS and the mutant control isoform SNACS^S785A were also stably transformed into *Arabidopsis*. Two independent transgenic lines were characterized for each of the two sensor constructs. SNACS fluorescence was observed throughout plant seedlings in the *35S:SNACS* expression lines, including guard cells and leaf epidermal cells (*Figure 2—figure supplement 1*). We tested effects of ABA on these sensors in stomatal guard cells. To conduct non-biased experiments, ABA or EtOH treatments were blinded in these assays and in these data analyses. Using a genetic background in which a tagged version of OST1 was expressed in the *OST1* T-DNA insertion allele *ost1-3* (*pUBQ10: OST1-HF/ost1-3*) (*Yoshida et al., 2002*; *Waadt et al., 2015*), application of 20 µM ABA induced a clear time-dependent increase in the FRET ratio, which reached stable saturation 2 to 4 min after application (*Figure 3A,C* to E, p=0.015, paired t-test, before ABA time point 3 min vs. after ABA time point 10 min; All following imaging analyses were evaluated by paired t-tests). Ethanol control

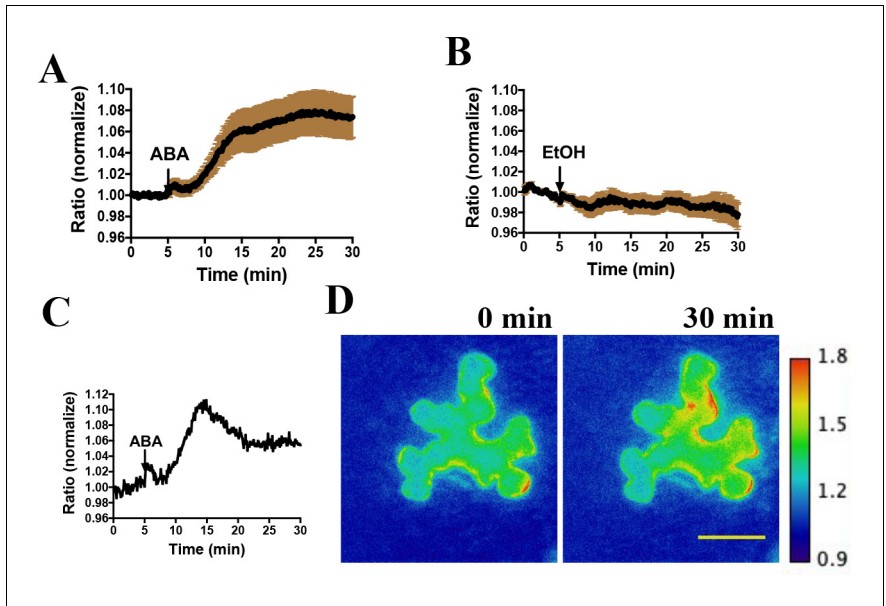

**Figure 2.** Abscisic acid induces increases in the ratio of YPet to Turquoise GL fluorescence emission of the SNACS reporter in *N. benthamiana*. (**A**) Time-resolved FRET ratio changes in response to ABA. SNACS was co-infiltrated with *pUBQ10:OST1-HF* in *N. benthamiana*. Leaf epidermises were perfused with assay buffer (5 mM KCl, 50 µM $CaCl_2$, 10 mM MES-Tris, pH 5.6) and then 20 µM ABA was added as indicated by the arrow. Experiments were repeated at least three times with similar results. FRET efficiency changes were recorded by measuring the ratio of fluorescence emissions at 535 nm/480 nm with an excitation wavelength of 434 nm (see Materials and methods). Data are averages of normalized emission ratios of YPet to Turquoise GL emission produced by exciting Turquoise GL from 11 cells. Error bars denote mean ± SEM. (**B**) Time-resolved FRET ratio in response to 0.02% EtOH (solvent control for ABA). Data are averages of normalized emission ratios from 7 cells. Error bars denote mean ± SEM. (**C**) and (**D**) Example of a single cell experiment from A. (**C**) The corresponding emission ratio normalized to the average value over 5 min before treatment. (**D**) Pseudo-colored fluorescence ratio images of SNACS-expressing *N. benthamiana* leaf epidermal cells at times 0 min and 30 min. The calibration bar to the right of (**D**) indicates the numerical ratio (non-normalized) scale corresponding to the heat map. Bar = 50 µm.

The online version of this article includes the following source data and figure supplement(s) for figure 2:

**Source data 1.** SNACS FRET ratio values from each stomate in *Figure 2*.
**Figure supplement 1.** Ratiometric image of 535/480 nm wavelength emissions show fluorescence in SNACS-expressing *Arabidopsis* epidermal cells (pseudo-colored fluorescence ratio scale is shown on the right, non-normalized).

---

applications showed no clear YPet/Turquoise GL emission ratio change (*Figure 3B*). Image analyses suggest that SNACS protein is either absent from the nucleus or is expressed at lower levels than in the cytoplasm, with fluorescence possibly 'bleeding through' from the cytoplasm to the nucleus (*Figure 3—figure supplement 1*).

Similar to the above experiments (*Figure 3*), in the Col-0 (wild-type) background, expression of SNACS showed ABA-induced emission ratio increases in guard cells (*Figure 4A and D*, n = 24 stomata, p=2 × $10^{-12}$ 3 min vs. 10 min time points). In additional experiments sets, expression of SNACS in the *pUBQ10:OST1-HF/ost1-3* background also showed ABA-induced emission ratio change (*Figure 4E and H*, n = 16 stomata, p=1.4 × $10^{-7}$ 3 min vs. 10 min). In transgenic lines expressing the mutant SNACS[S785A], that corresponds to the Ser-30-Ala mutation in AKS1, no clear increases in the ratio of YPet to Turquoise GL emission ratio were observed after ABA application (*Figure 4C and G*; n = 9 stomata (4C), p=0.892, 3 min vs. 10 min; n = 12 stomata (4G), p=0.184, 3 min vs. 10 min). These data indicate that the phosphorylation site Ser 785 residue of SNACS (*Figure 4C and G*) is necessary for ABA-induced increases in the ratio of YPet to Turquoise GL emission in plant cells. In additional EtOH application controls, both in SNACS and in SNACS[S785A] - expressing transgenic lines no substantial EtOH-induced increases were observed in the ratio of YPet to Turquoise GL emission (*Figure 4B,F* and *Figure 4—figure supplement 1*; n = 9 stomata

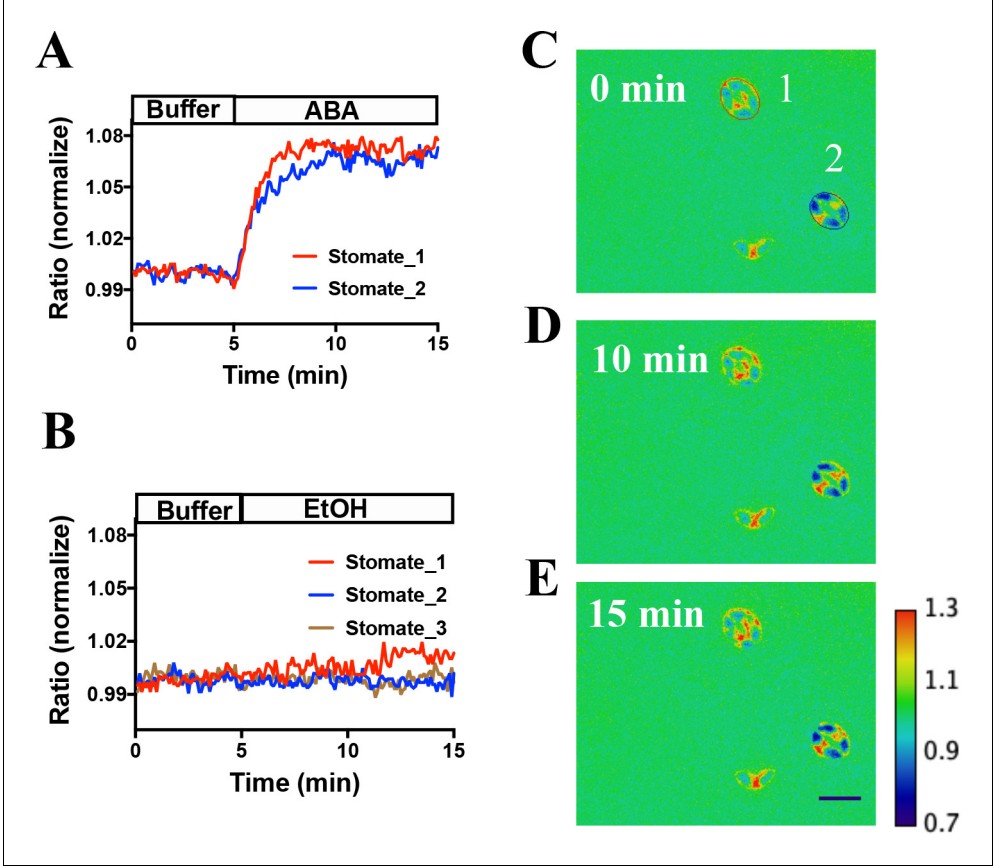

**Figure 3.** In vivo response of SNACS reporter of SnRK2 activity in response to ABA in *Arabidopsis* guard cells. (A) Time-resolved FRET ratio changes in response to ABA (20 μM). FRET efficiency changes were recorded by measuring the ratio of fluorescence emissions at 535 nm/480 nm with an excitation wavelength of 434 nm (see Materials and methods). (B) Time-resolved FRET ratio change in response to EtOH (0.02%, solvent control for ABA). (C to E) Images of SNACS fluorescence ratios in guard cells from A at 0 min, 10 min and 15 min. The colored circles indicate the regions of interest (ROIs) used to measure fluorescence emissions with the colors corresponding to the blue and red traces in panel A. '1' and '2' in (C) denote 'stomate_1' and 'stomate_2', respectively. The calibration bar in the lower right of (E) indicates the numerical scale corresponding to the non-normalized heat map. Bar = 20 μm. Note that the lowest cell (red spot) in the images shows fluorescence of a single guard cell from an apparent half stomate. Treatments were performed blinded (ABA or EtOH). SNACS FRET activities in guard cells in leaf epidermises were analyzed in a *pUBQ10:OST1-HF*-expressing *ost1-3* genetic background (*Waadt et al., 2015*). The ratio of YPet to Turquoise GL emission was normalized to the average value over 5 min before treatment.

The online version of this article includes the following source data and figure supplement(s) for figure 3:

**Source data 1.** SNACS FRET ratio values from each stomate in *Figure 3*.

**Figure supplement 1.** SNACS fluorescence appears to be observed mainly in the cytoplasm, with possible low expression in the nucleus or fluorescence may 'bleed' from the cytoplasm towards the nuclear region.

(4B), p=0.808, 3 min vs. 10 min; n = 11 stomata (4F), p=0.112, 3 min vs. 10 min). Average time-dependent ratiometric fluorescence changes for data including data in *Figure 4* are shown in *Figure 4—figure supplement 2A–H*.

We next investigated whether the SNACS ratio change was dependent on SnRK2 activity in planta. As SnRK2.2, SnRK2.3 and SnRK2.6 have been shown to contribute to ABA signaling in guard cells (*Takahashi et al., 2017*; *Fujii and Zhu, 2009*; *Brandt et al., 2015*; *Takahashi and Kinoshita, 2015*), we examined SNACS in Col-0, *ost1/snrk2.6* single mutant, *snrk2.2/2.3* double mutant and *snrk2.2/2.3/2.6* triple mutant plants (*Figure 5*). Interestingly, ABA caused clear fluorescence emission ratio increases in *ost1/snrk2.6* guard cells (*Figure 5B*; n = 14 stomata, p=3.2 × 10$^{-7}$ 3 min vs. 10 min). Compared to the Col-0 guard cells, in the *ost1/snrk2.6* background ABA induced slower

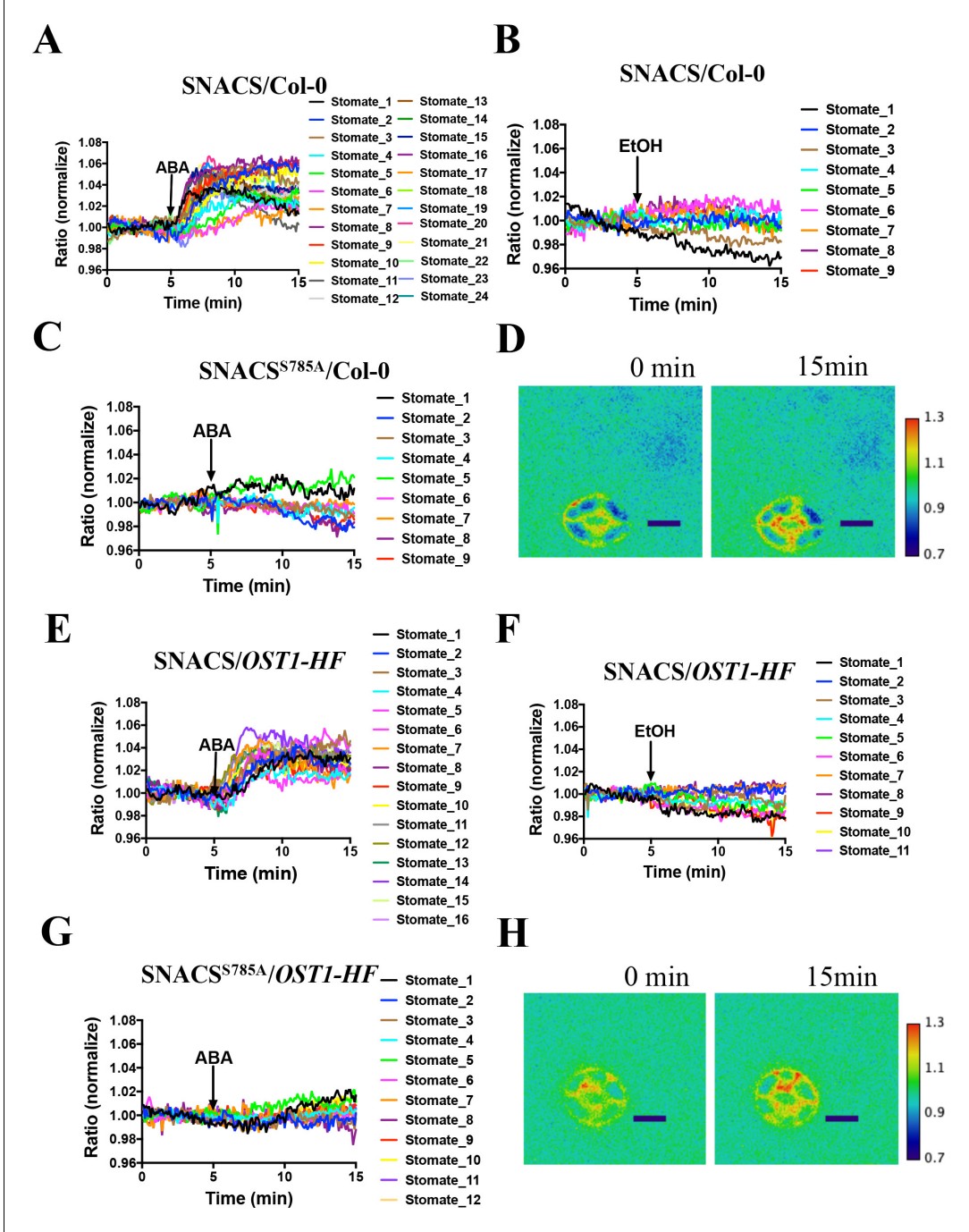

**Figure 4.** The amino acid AKS1 Ser-30 in the SNACS reporter is necessary for ABA-induced ratio increases in *Arabidopsis* guard cells. Leaf epidermises from transgenic lines expressing SNACS were used for depicting the ratio of YPet to Turquoise GL emission produced by exciting Turquoise GL with 434 nm light. (**A**) ABA (20 µM) induced the SNACS response in the Col-0 genetic background. (**B**) SNACS response to EtOH (0.02%, solvent control for ABA) treatment in the Col-0 wild-type genetic background. (**C**) The S785A mutation in the SNACS reporter impairs ABA-induced FRET ratio changes in the Col-0 wild-type genetic background. (**D**) A representative SNACS fluorescence ratio image from A at 0 min and 15 min time points. Bar = 10 µm. (**E**) ABA (20 µM) induced SNACS response in the *pUBQ10:OST1-HF*-expressing in *ost1-3* genetic background. (**F**) SNACS response to EtOH (0.02%, solvent control for ABA) in *pUBQ10:OST1-HF*- expressing in *ost1-3* genetic background. (**G**) The S785A mutation in the SNACS impairs ABA-induced FRET ratio changes in the *pUBQ10:OST1-HF*-expressing in *ost1-3* genetic background. The ratios were normalized to the average value over the 5 min before treatment. (**H**) A representative SNACS fluorescence ratio image from E at 0 min and 15 min time points. Bar = 10 µm. Calibration bars to the right of D and H show the numerical ratio (non-normalized) scale corresponding to the pseudo-coloring.

The online version of this article includes the following source data and figure supplement(s) for figure 4:

*Figure 4 continued on next page*

*Figure 4 continued*

**Source data 1.** SNACS FRET ratio values from each stomate in *Figure 4*.
**Figure supplement 1.** EtOH controls for SNACS^S785A in Col-0 and in *pUBQ10:OST1-HF* expressed in the *ost1-3* genetic background.
**Figure supplement 1—source data 1.** SNACS FRET ratio values from each stomate in *Figure 4—figure supplement 1*.
**Figure supplement 2.** Combined and averages of single stomate imaging data from experimental sets including experiments in *Figure 4*.
**Figure supplement 2—source data 1.** SNACS FRET ratio values from each stomate in *Figure 4—figure supplement 2*.

emission ratio increases (*Figure 5A and B*; n = 12 stomata (5A), p=2.4 × 10^{-6} 3 min vs. 10 min). In *snrk2.2/2.3* double mutant guard cells, ABA induced FRET ratio increases were observed (*Figure 5C*, n = 20 stomata (5C), p=5.7 × 10^{-9} 3 min vs. 10 min). Notably, no ABA-induced ratio increase was observed in the *snrk2.2/2.3/2.6* triple mutant guard cells, which suggests that SnRK2

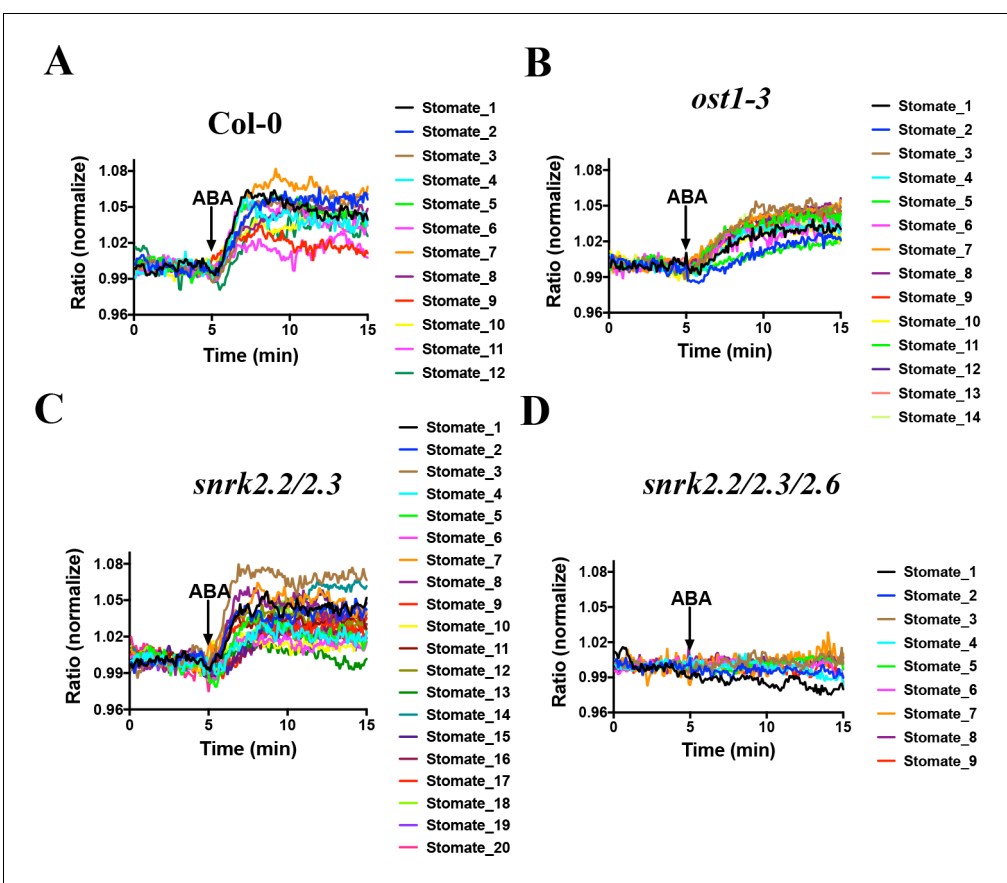

**Figure 5.** SnRK2 activity is needed in guard cells for ABA-induced increases in the FRET ratio of SNACS. Leaf epidermises from transgenic lines expressing SNACS were used for analyzing the ratio of YPet to Turquoise GL emission in guard cells produced by exciting Turquoise GL with 434 nm light. The ratio was normalized to the average value over the 5 min before ABA exposure. (A) ABA (20 μM) induced SNACS responses in guard cells of the Col-0 (WT) genetic background. (B) ABA (20 μM) induced SNACS responses in guard cells of the *ost1-3* genetic background. (C) ABA (20 μM) induced SNACS responses in guard cells of the *snrk2.2/2.3* double mutant genetic background. (D) ABA (20 μM) induced SNACS response was impaired in *snrk2.2/2.3/2.6* triple mutant guard cells.

The online version of this article includes the following source data and figure supplement(s) for figure 5:

**Source data 1.** SNACS FRET ratio values from each stomate in *Figure 5*.
**Figure supplement 1.** Combined and averages of single stomate imaging data from experimental sets shown in *Figure 5*.
**Figure supplement 1—source data 1.** SNACS FRET ratio values from each stomate in *Figure 5—figure supplement 1*.

activity is required for ABA-induced SNACS-dependent FRET ratio changes in planta (*Figure 5D*, n = 9 stomata, p=0.122 3 min vs. 10 min). Average time-resolved fluorescence emission data from *Figure 5* are shown in *Figure 5—figure supplement 1*. These data are consistent with findings that all three SnRK2s contribute to ABA signaling in guard cells and that the triple mutant has the strongest physiological phenotype (*Fujii and Zhu, 2009*; *Brandt et al., 2015*). Taken together, these data provide strong evidence that time-dependent SnRK2 activity can be detected in single live cells by SNACS both in *N. benthamiana* and *Arabidopsis*.

## Effects of kinase inhibitors on SNACS

Previous research has led to the hypothesis that basal ABA signaling and basal SnRK2 activity occur in non-stressed guard cells (*Hsu et al., 2018*; *Yoshida et al., 2019*; *Lahr and Raschke, 1988*). However, experimental evidence directly examining the proposed basal activity of SnRK2 kinases in intact guard cells is lacking. Experiments were pursued to determine the effects of protein kinase inhibition during SNACS recordings. In order to determine the effects of kinase inhibition, the general Ser/Thr protein kinase inhibitor, K252a was used, which abolishes ABA-induced stomatal closing (*Schmidt et al., 1995*). We found that 10 μM K252a application resulted in a time-resolved drop in the FRET emission ratio in *Arabidopsis* guard cells (*Figure 6A*; n = 13 stomata (6A), p=4.9 × 10$^{-6}$ 3 min vs.10 min; n = 11 stomata (6B, controls), p=0.158 3 min vs. 10 min). In addition, ABA did not induce measurable emission ratio increases after K252a application, suggesting that ABA-induced emission ratio increases are caused by K252a-sensitive protein kinase activity in vivo (*Figure 6A*). Next, we added 10 μM K252a following 10 min ABA treatments. Interestingly, after ABA treatment and subsequent K252a exposure, the FRET emission ratio decreased, suggesting that SNACS can reversibly report protein kinase activity (*Figure 6C*; n = 10 stomata (6C), p=0.0004 3 min vs.10 min, p=0.0001 10 min vs. 20 min; n = 8 stomata (6D, controls), p=0.00003 3 min vs.10 min, p=0.836 10 min vs. 20 min).

As control, we tested whether the calmodulin inhibitor W7 affects SnRK2 protein kinase activity in plant cells. *In-gel* kinase assays showed that W7 does not have a clear effect on ABA-induced OST1/SnRK2.6 activation in *Arabidopsis* mesophyll cell protoplasts (*Figure 6—figure supplement 1*). In controls K252a inhibited ABA-induced *in-gel* kinase activity (*Figure 6—figure supplement 1*). For comparison to the K252a inhibitor, we investigated the effect of W7 (*Rudd et al., 1996*) on the SNACS reporter in guard cells. We found no FRET ratio changes upon W7 treatment (*Figure 6E and F*; n = 8 stomata (6E), p=0.988 3 min vs.10 min; n = 7 stomata (6F, controls), p=0.348 3 min vs.10 min). After W7 exposure, application of ABA caused ABA-induced FRET shifts (*Figure 6E and F*; p=1.1 × 10$^{-9}$ (6E), p=7.4 × 10$^{-6}$ (6F), 10 min vs. 20 min). Average time-resolved fluorescence emission ratio changes are shown in *Figure 6—figure supplement 2* that include the data in *Figure 6*. Taken together, these data suggest that SNACS is a reversible protein kinase activity reporter and that SNACS can report basal SnRK2 activity in vivo, that is down-regulated upon kinase inhibition.

## Methyl-Jasmonate does not induce SNACS FRET ratio changes in guard cells

Methyl jasmonate (MeJA) has been previously reported to induce stomatal closure (*Herde and Pena-Cortes, 1997*; *Gehring, 1997*; *Suhita et al., 2004*; *Akter et al., 2012*). The OST1/SnRK2.6 kinase is required for MeJA-induced stomatal closure (*Yin et al., 2016*). Therefore, next we analyzed the effect of MeJA on SNACS FRET changes in guard cells. Exogenous application of MeJA did not result in robust FRET emission ratio changes in *Arabidopsis* guard cells (*Figure 7*, *Figure 7—figure supplement 1*; n = 9 stomata (7A), p=0.981 3 min vs. 15 min, p=0.896 3 min vs. 25 min; n = 12 stomata (7B, controls), p=0.553 3 min vs. 15 min, p=0.664, 3 min vs. 25 min). Furthermore, in the presence of MeJA, subsequent application of ABA caused increases in the SNACS emission ratio in these experiments (*Figure 7*).

## CO$_2$ elevation does not induce SNACS FRET ratio increases in guard cells

*snrk2.6/ost1* mutant alleles show impaired and slowed CO$_2$-induced stomatal closure (*Xue et al., 2011*; *Merilo et al., 2013*; *Hsu et al., 2018*). Interestingly, a recent study suggests that CO$_2$ elevation does not activate SnRK2.6/OST1 protein kinases using *in-gel* kinase assays with isolated

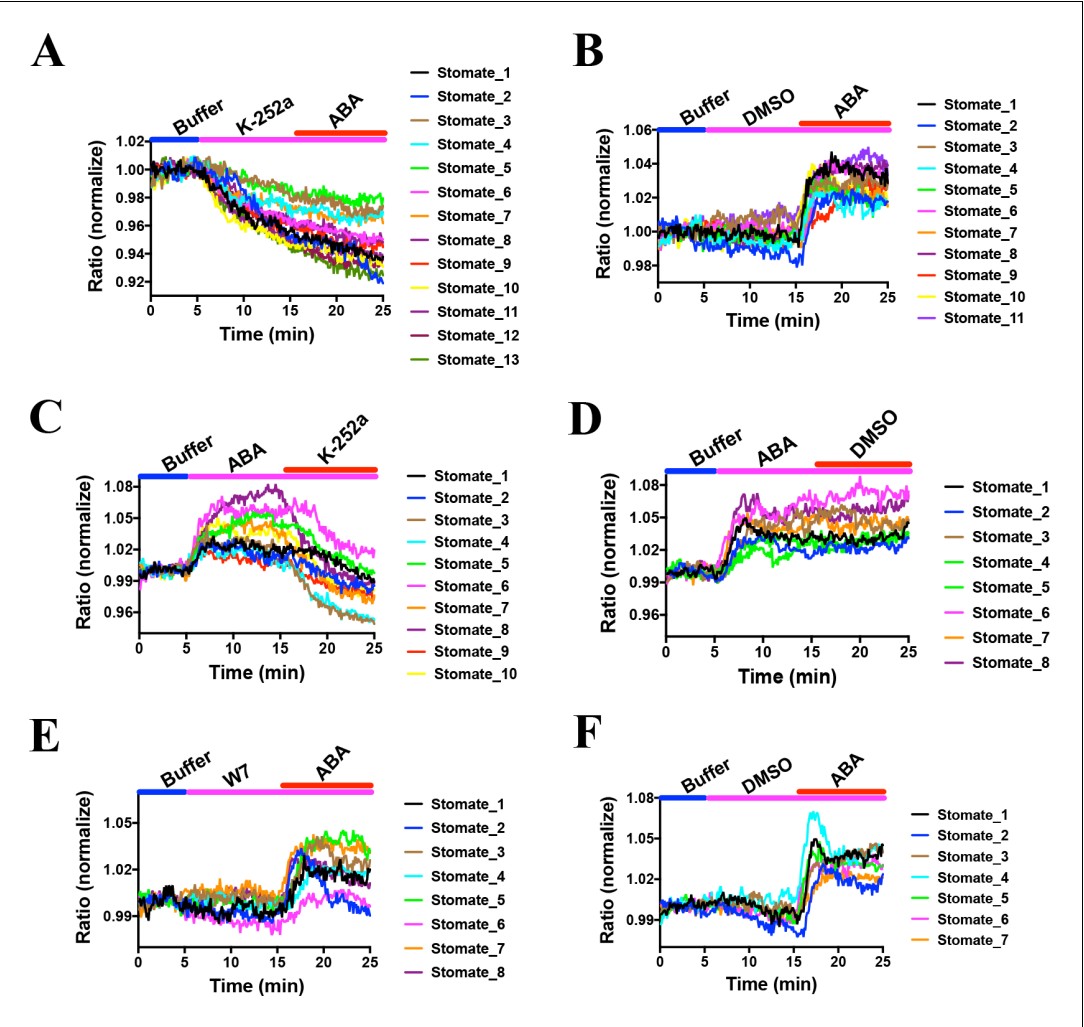

**Figure 6.** Effects of kinase inhibitors on SNACS fluorescence emission ratios in guard cells. SNACS responses in guard cells were analyzed in the *pUBQ10:OST1-HF*-expressed in the *ost1-3* genetic background. The ratio of YPet to Turquoise GL emission was normalized to the average value over the 5 min before K252a application. (**A**) The protein kinase inhibitor K-252a reduced SNACS FRET ratio in vivo, and ABA did not induce a ratio increase in the presence of K-252a. After 10 min incubation with 10 μM K-252a, 20 μM ABA was added. (**B**) ABA induced a ratio increase in the presence of DMSO (0.2%, solvent control for K-252a). (**C**) The kinase inhibitor K-252a inhibited SnRK2 kinase activity after ABA treatment. (**D**) Control experiment for C. 0.2% DMSO was added. (**E**) Kinase inhibitor W7 did not affect SNACS FRET ratio in vivo. After 10 min incubation with 20 μM W7, 20 μM ABA was added. (**F**) Control experiment for E. 0.1% DMSO was added as solvent for W7.

The online version of this article includes the following source data and figure supplement(s) for figure 6:

**Source data 1.** SNACS FRET ratio values from each stomate in *Figure 6*.

**Figure supplement 1.** W-7 does not have a clear effect on ABA-induced OST1/SnRK2.6 activation in plant cells.

**Figure supplement 1—source data 1.** Uncropped gel images for *Figure 6—figure supplement 1*.

**Figure supplement 2.** Combined and averages of single stomate imaging data from experimental sets including experiments in *Figure 6*.

**Figure supplement 2—source data 1.** SNACS FRET ratio values from each stomate in *Figure 6—figure supplement 2*.

---

*Arabidopsis* guard cell protoplasts (*Hsu et al., 2018*). However, based on the present debate whether $CO_2$ elevation directly activates early ABA signal transduction (*Dittrich et al., 2019*) or not (*Hsu et al., 2018*; *Merilo et al., 2013*), real-time analyses in intact guard cells are needed. Here, using SNACS-expressing *Arabidopsis* plants, we tested whether changes in $CO_2$ concentration affect SNACS FRET activity in intact guard cells in real-time analyses. Intact leaf epidermises (*Young et al.,*

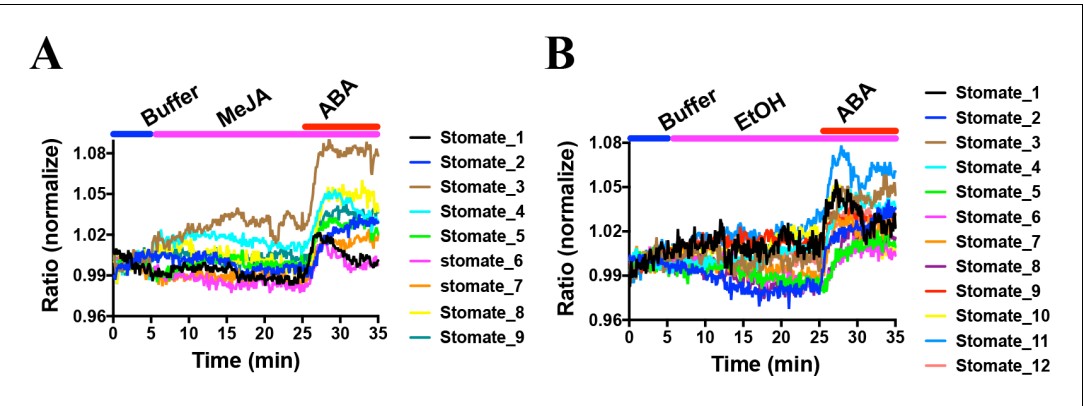

**Figure 7.** MeJA did not induce consistent FRET ratio changes of the SNACS SnRK2 activity reporter in guard cells under the imposed conditions. The SNACS reporter was analyzed in guard cells in leaf epidermises of plants expressing *pUBQ10:OST1-HF* in the *ost1-3* genetic background. The ratio of YPet to Turquoise GL emission was normalized to the average value over the 5 min before treatment. (**A**) MeJA (20 µM) did not affect the FRET ratio in guard cells. ABA induced ratio increases in the presence of MeJA. After 20 min treatment with 20 µM MeJA, 20 µM ABA was added. (**B**) Control experiment for A. 0.1% EtOH (solvent control for MeJA) was added.

The online version of this article includes the following source data and figure supplement(s) for figure 7:

**Source data 1.** SNACS FRET ratio values from each stomate in *Figure 7*.

**Figure supplement 1.** Combined and averages of single stomate imaging data from experimental sets including experiments in *Figure 7*.

**Figure supplement 1—source data 1.** SNACS FRET ratio values from each stomate in *Figure 7—figure supplement 1*.

---

*2006*) were incubated in a low $CO_2$ buffer (115 ppm) for 10 min, and the buffer was replaced with a high $CO_2$ buffer (1170 ppm) by perfusion while monitoring SNACS FRET ratios. FRET emission ratios showed no measurable increases after a 40 min incubation with the high $CO_2$ buffer (*Figure 8A* and *Figure 8—figure supplement 1A*, n = 9 stomata, p=0.184 3 min vs. 15 min). After high $CO_2$ treatment, 20 µM ABA was added to the buffer. We observed rapid FRET ratio increases in response to ABA (*Figure 8A*). Low $CO_2$ causes a rapid opening of stomatal pores. We further tested whether low $CO_2$ disrupts ABA-induced FRET ratio changes of the SNACS reporter. No clear FRET ratio increases were detected for continuous 30 min low $CO_2$ exposure (115 ppm). Subsequent exposure to 20 µM ABA caused ratio increases (*Figure 8B* and *Figure 8—figure supplement 1B*, n = 14 stomata, p=$5.9 \times 10^{-6}$ 20 min vs. 35 min).

$CO_2$-induced stomatal closing is >80% completed within 15 to 20 min of a $CO_2$ concentration elevation in wild-type (Col-0) *Arabidopsis* (*Young et al., 2006*). In previous research using the ABA nano-reporter ABAleon2.15, a shift in the $CO_2$ concentration from 115 ppm to 535 ppm did not cause a measurable increase in the ABA concentration in guard cells, even though ABA receptors and OST1 were found to be important for the $CO_2$ response (*Hsu et al., 2018*), which was differently interpreted in a recent study (*Dittrich et al., 2019*). Nevertheless, we pursued experiments here to determine whether a shift to a higher $CO_2$ concentration of 1170 ppm could cause a rapid increase in the ABA concentration in guard cells. No dramatic ABAleon2.15 emission ratio change was observed upon increasing the $CO_2$ concentration from 115 ppm to 1170 ppm (*Figure 8C* and *Figure 8—figure supplement 1C*, n = 10 stomata, p=0.502 3 min vs. 30 min). At the end of these experiments, intact abaxial leaf epidermises were exposed to 20 µM ABA, which caused a rapid reduction in the ABAleon2.15 FRET ratio (*Figure 8C,D* and *Figure 8—figure supplement 1C*), which corresponds to an increase in cellular ABA concentration (*Waadt et al., 2014*). In control experiments, no detectable emission ratio change was observed in 30 min of low $CO_2$ buffer exposure (115 ppm), followed by a rapid reduction in ratio upon exposure to 20 µM ABA (*Figure 8D* and *Figure 8—figure supplement 1D*; n = 9 stomata, p=$6.1 \times 10^{-7}$ 20 min vs. 35 min). Thus, no clear rapid increase in the cellular ABA concentration of guard cells was found within the time frame in which $CO_2$-induced stomatal closing occurs.

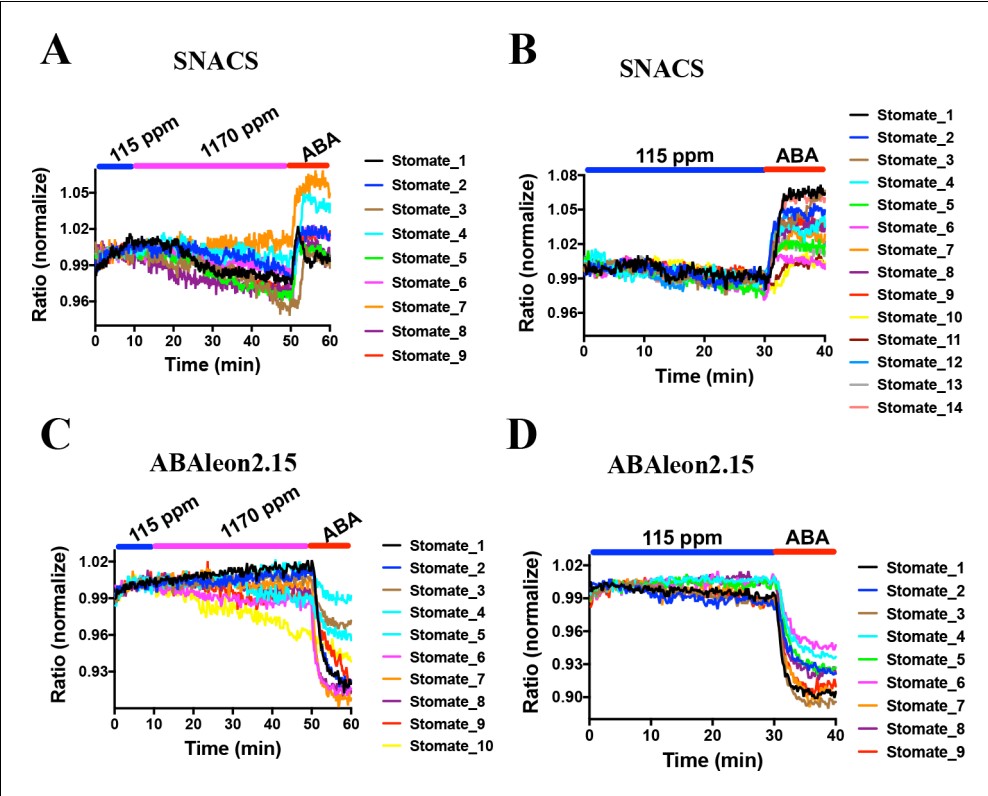

**Figure 8.** $CO_2$ elevation neither induces SNACS FRET ratio increases nor affects ABA concentration in guard cells. Fluorescence ratios of guard cells in intact leaf epidermises were analyzed in the *pUBQ10:OST1-HF* expressed in the *ost1-3* genetic background (**A** and **B**). The ratio of YPet to Turquoise GL emission was normalized to the average value over the 10 min before treatments. (**A**) High $CO_2$ (1170 ppm) did not increase the FRET ratio in guard cells. Intact leaf epidermises were exposed to low $CO_2$ (115 ppm), then switched to high $CO_2$ (1170 ppm) buffer, and then exposed to 20 µM ABA containing buffer. (**B**) Leaf epidermises were exposed to low $CO_2$ (115 ppm), then exposed to 20 µM ABA containing buffer. (**C**) Time-resolved ABAleon2.15 emission ratios in guard cells in response to $CO_2$ changes and ABA application. Intact leaf epidermises were exposed to low $CO_2$ (115 ppm), then switched to high $CO_2$ (1170 ppm) buffer, and then exposed to 20 µM ABA containing buffer. (**D**) Leaf epidermises were exposed to low $CO_2$ (115 ppm), then exposed to 20 µM ABA containing buffer. Note that reduction in fluorescence ratios observed in ABAleon2.15-expressing guard cells correspond to ABA concentration increases (*Waadt et al., 2014*).

The online version of this article includes the following source data and figure supplement(s) for figure 8:

**Source data 1.** SNACS FRET ratio values from each stomate in *Figure 8*.

**Figure supplement 1.** Combined and averages of single stomate imaging data from experimental sets including experiments in *Figure 8*.

**Figure supplement 1—source data 1.** SNACS FRET ratio values from each stomate shown in *Figure 8—figure supplement 1*.

## PYL4 and PYL5 ABA receptors are not essential for $CO_2$-induced stomatal closing

A recent study has suggested that $CO_2$-induced stomatal closing requires PYL4 and PYL5 ABA receptors (*Dittrich et al., 2019*). As the present study has found no clear $CO_2$ activation of SnRK2s in guard cells and no $CO_2$-induced ABA increases in guard cells, we pursued gas exchange experiments in intact plants with mutants lacking *PYL4* and *PYL5* expression. Experiments were conducted on higher order ABA receptor mutants that have been shown to disrupt ABA-induced stomatal closing (*Merilo et al., 2018*; *Gonzalez-Guzman et al., 2012*). Experiments were performed in quintuple ABA receptor mutants: *pyr1 pyl1 pyl4 pyl5 pyl8* ('pyl-11458') and *pyr1 pyl2 pyl4 pyl5 pyl8* ('pyl-12458') and sextuple mutant plants: *pyr1 pyl1 pyl2 pyl4 pyl5 pyl8* ('pyl-112458').

Time-resolved gas exchange experiments measuring whole intact plants showed that all tested plant lines displayed stomatal closure in response to changing the $CO_2$ concentration from ambient (400 ppm) to 800 ppm (*Figure 9A and B*). Earlier studies have shown the crucial role of PYR/RCAR receptors in maintaining steady-state stomatal conductance (*Hsu et al., 2018*; *Gonzalez-Guzman et al., 2012*; *Merilo et al., 2013*). ABA receptor sextuple mutant plants displayed 5.2 times higher pre-$CO_2$-treatment stomatal conductance compared to wild-type (*Figure 9A and C*). ABA receptor quintuple *pyl-11458* and *pyl-12458* mutants maintained 2.3 and 2.9 times higher pretreatment stomatal conductances compared to wild-type (*Figure 9A and C*). After application of elevated $CO_2$, the stomatal conductance of all ABA receptor mutant lines decreased to a similar or even higher extent than in wild-type plants even though their stomatal conductances remained higher after 50 min in elevated $CO_2$ (*Figure 9A and C*). This can be in part explained by an increased stomatal density in higher order ABA receptor knockout mutants and a weaker response to basal ABA concentrations (*Hsu et al., 2018*; *Merilo et al., 2018*). Stomatal responses to $CO_2$ shifts from 400 to 800 ppm were clearly delayed in PYR/RCAR sextuple mutant plants as indicated by a roughly twice longer half-response time (*Figure 9D*), consistent with previous findings (*Hsu et al., 2018*). We also calculated reductions in stomatal conductance after 20 min under elevated $CO_2$ as an approximation for the initial phase of the $CO_2$ responses (*Figure 9—figure supplement 1*). Such analyses showed that in absolute units PYR/RCAR quintuple mutant *pyl-12458* and sextuple mutant *pyl-112458* plants displayed a larger absolute reduction in stomatal conductance in response to high $CO_2$ than wild-type plants.

To further investigate the role of PYL4 and PYL5 in $CO_2$-induced stomatal closing, we pursued a light protocol followed by $CO_2$ elevation, as in experiments used by *Dittrich et al., 2019*. In these experiments, *pyl-11458* quintuple mutant leaves showed stomatal opening in response to 125 μmol m$^{-2}$ s$^{-1}$ light as expected. After light-induced stomatal opening, $CO_2$ elevation showed similar $CO_2$-induced stomatal closing in the wild-type (Col-0) background as in the *pyl-11458* quintuple mutant that lacks *PYL4* and *PYL5* (*Figure 9—figure supplement 2A*). In complementation lines expressing *PYL4*, *PYL5* or *PYL1* under control of the guard cell *pGC1* promoter, no clear enhancement of the final $CO_2$ response was observed (*Figure 9—figure supplement 2B* to D). Thus in the two laboratories in which $CO_2$-regulation of stomatal conductance was investigated in intact plants in the present study (EM and HK lab and JS lab), we could not confirm that the absence of PYL4 and PYL5 disrupts the stomatal $CO_2$ response (*Dittrich et al., 2019*).

## Discussion

In plants, SnRK2 protein kinases play critical roles in abiotic stress responses (*Boudsocq and Laurière, 2005*; *Umezawa et al., 2009*; *Nakashima et al., 2009*; *Fujita et al., 2009*; *Fujii and Zhu, 2009*; *Li et al., 2000*; *Yoshida et al., 2002*; *Mustilli et al., 2002*). Abscisic acid activation of SnRK2 protein kinases is required for ABA signal transduction (*Cutler et al., 2010*; *Raghavendra et al., 2010*; *Zhu, 2016*). However, the common, *in-gel* SnRK2 kinase assay method has significant limitations for monitoring cell type, cellular compartment and time-dependent activation of these protein kinases. *In-gel* kinase assays in *Arabidopsis* guard cells require purification of 10$^5$ or more guard cell protoplasts from more than 100 leaves and 10 plants for each individual guard cell *in-gel* kinase assay lane. Real-time FRET measurements of SnRK2 protein kinase activities in intact individual plant cells and tissues using the SNACS reporter enable direct investigation of key biological questions in real time, as shown here.

In this study, we developed a SnRK2 protein kinase sensor, SNACS, containing an AKS1 substrate domain and a full length 14-3-3 protein. The SnRK2.6/OST1 kinase phosphorylated the SnRK2 sensor in dependence of the AKS1 Ser-30 residue in vitro (*Figure 1C*), which is essential for 14-3-3 protein binding in ABA signal transduction (*Takahashi et al., 2013*). In vitro and in planta analyses show an increase in FRET ratio in a SnRK2 kinase activity-dependent manner (*Figures 1D,E* and *5*). We further observed ABA-induced FRET ratio increases in *Nicotiana benthamiana* epidermal cells and *Arabidopsis* guard cells (*Figures 2* and *3*). Phosphorylation of the AKS1 Ser-30 residue is required for 14-3-3 protein binding to the AKS1 protein (*Takahashi et al., 2013*). Notably, the mutant AKS1 Ser-30-Ala sensor isoform does not show FRET ratio changes both in vitro and in plant cells (*Figures 1F* and *4*). These results are consistent with a model in which changes in FRET ratio are derived from a conformational change through 14-3-3 protein binding to the phosphorylated AKS1 domain in

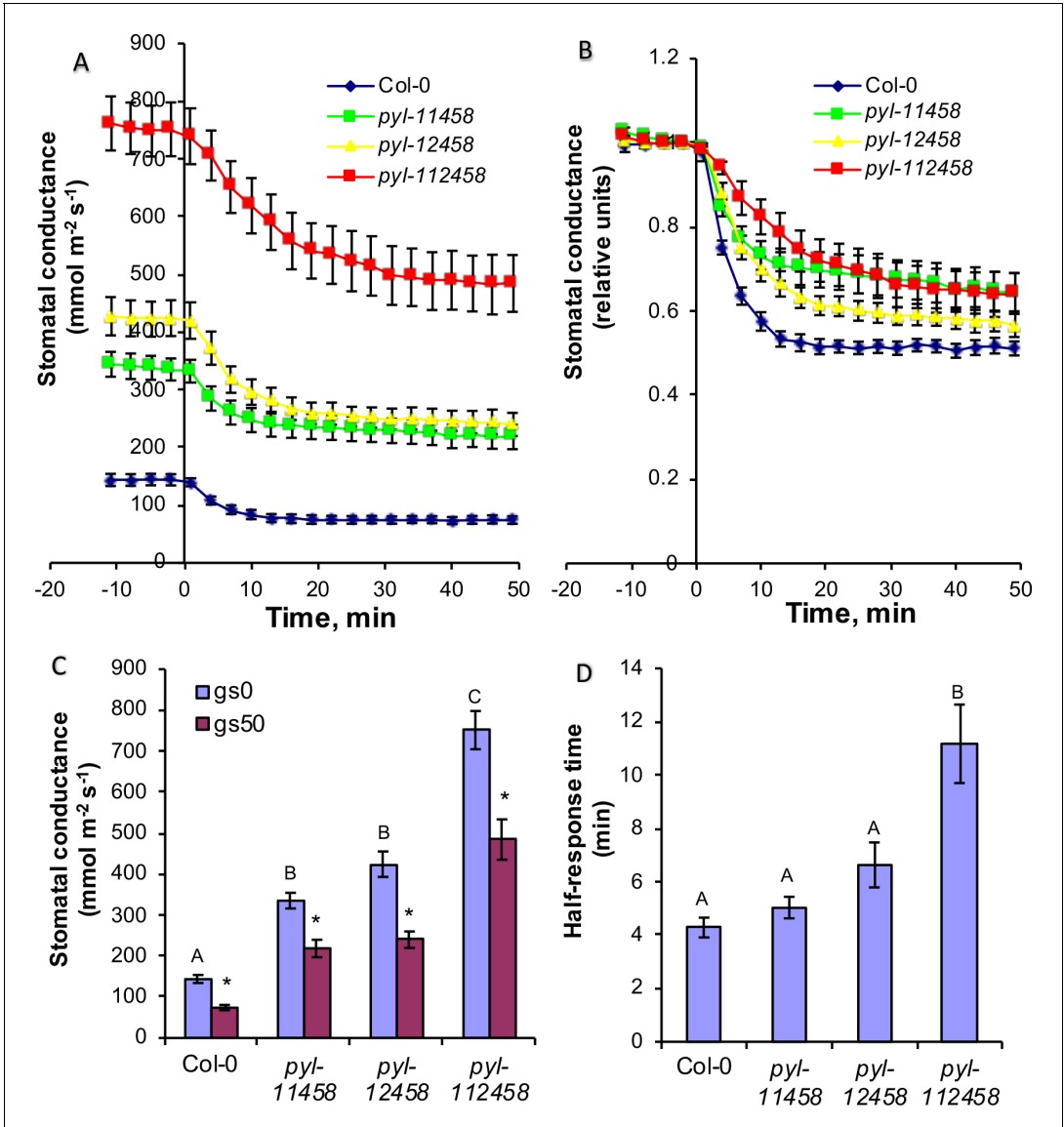

**Figure 9.** $CO_2$ elevation causes robust stomatal closing responses in two ABA receptor quintuple mutant and in ABA receptor sextuple mutant plants. (A, B) Time-resolved stomatal conductances of PYR/RCAR receptor quintuple (*pyl-11458* and *pyl-12458*) and sextuple (*pyl-112458*) mutants. Air $CO_2$ concentration was increased from 400 ppm to 800 ppm at the zero timepoints. Gas exchange analyses are from n = 5 to 7 whole plants per genotype and condition, using methods described in *Merilo et al., 2018* (see Materials and methods). Data are presented in absolute (A) and relative (B) units. (C) Stomatal conductances before (gs0) and 50 min (gs50) after changing $CO_2$ from 400 ppm to 800 ppm, average and SE (n = 5 to 7 whole plants). Capital letters denote differences in gs0 (pre-treatment stomatal conductance) between the lines (n = 5 to 7 whole plants, ANOVA, p=0.000000), whereas stars denote whether gs50 is significantly different from gs0 of that line (n = 5 to 7 whole plants, ANOVA, p=0.00001). (D) Half-response times of stomatal closures in response to 50 min $CO_2$ enrichment. Capital letters denote significant differences between the lines (n = 5 to 7 whole plants, ANOVA, p=0.000023).

The online version of this article includes the following source data and figure supplement(s) for figure 9:

**Source data 1.** Stomatal conductance values of individual plants and half response times.

**Figure supplement 1.** PYR/RCAR ABA receptor *pyr1 pyl2/4/5/8* quintuple mutant (*pyl-12458*) and *pyr1 pyl1/2/4/5/8* (*pyl-112458*) sextuple mutant show larger absolute stomatal closing responses during the first 20 min under elevated $CO_2$ compared to wild-type (Col-0) (light blue bars).

**Figure supplement 1—source data 1.** Absolute and relative changes in stomatal conductance values used in *Figure 9—figure supplement 1*.

**Figure supplement 2.** Stomata of ABA receptor quintuple mutant (*pyr1/pyl1/pyl4/pyl5/pyl8*) and guard cell-targeted (B) PYL1 and (C) PYL4 ABA receptor complemented plants show stomatal $CO_2$ responses similar to wild-type (Col-0) controls.

**Figure supplement 2—source data 1.** Stomatal conductance values of individual plants used in *Figure 9—figure supplement 2*.

**Figure supplement 3.** Genotyping of ABA receptor multiple mutants and guard cell-targeted ABA receptor complemented plants.

SNACS (*Figure 1B*). The full length AKS1 protein functions as a transcription factor in the nucleus. However, the bHLH domain of AKS1, which is important for its transcription factor function and nuclear localization, was excluded from our SNACS sensor construct. SNACS fluorescence appears to be observed mainly in the cytoplasm and in cytoplasmic spaces between vacuoles, with possible weak or no expression in the nucleus (*Figure 3—figure supplement 1*). Future research can target SNACS to the nucleus or to other cellular compartments.

ABA caused relatively small SNACS FRET ratio shifts in guard cells. Blind ABA vs. EtOH control treatment experiments and controls with the mutant SNACS[S785A] and *snrk2.2/2.3/2.6* triple mutant guard cells confirmed the ABA response of SNACS (*Figures 3*, *4* and *5*). Protein kinase sensors for protein kinases have been developed that show small ratio shifts of 1% to 4% and have been used to address key biological questions (e.g.*Kajimoto et al., 2019*). In our study, the average response of SNACS to ABA was 3.7 ± 0.2% (n = 46 stomata,± SEM).The low background noise in these kinase sensors has enabled real-time analyses of kinase activation and inhibition (*Kajimoto et al., 2019*). Therefore, because of small ratio shifts, continuous recordings in individual cells are needed to resolve stimulus-dependent FRET changes (*Allen et al., 1999*), rather than investigating single time point comparisons of FRET ratios in different cells. FRET imaging with SNACS permits single cell-type imaging in real-time, compared to the need to purify very large numbers of cells and isolate extracts for each *in-gel* kinase assay lane.

The protein kinase inhibitor K252a, that is known to disrupt ABA signaling in guard cells (*Schmidt et al., 1995*), inhibits the ABA-induced FRET increase (*Figure 6A*). Furthermore, *snrk2.2/2.3/2.6* triple mutants showed no clear ABA-induced FRET ratio shift (*Figure 5D*). These data suggest that the ABA-induced FRET ratio changes are dependent on protein phosphorylation (*Figure 1C*). Interestingly, we also found that K-252a induces a decrease of the SNACS FRET ratio when K252a was applied after the ABA-induced FRET ratio increase (*Figure 6C*). This would be consistent with a dephosphorylation of SNACS by a protein phosphatase in plant cells. We do not exclude the possibility that K252a affects another protein kinase activity, in addition to SnRK2 activities (*Figure 6—figure supplement 1*). The calmodulin inhibitor, that does not inhibit SnRK2.6/OST1 did not affect SNACS fluorescence ratios (*Figure 6E*; *Figure 6—figure supplement 1*). In *ost1/ snrk2.6* single mutant guard cells and in *snrk2.2/snrk2.3* double mutant guard cells, ABA caused increases in the FRET ratio of SNACS, but not in *snrk2.2/2.3/2.6* triple mutant guard cells (*Figure 5*). Furthermore, SNACS-dependent FRET signals were reversed by protein kinase inhibitors (*Figure 6*). Together these data suggest that SNACS can report low baseline SnRK2 activity, and that SNACS can dynamically report biological changes in activity. HAB1 (also named AtP2C-HA) (*Chérel et al., 2002*; *Saez et al., 2004*; *Leonhardt et al., 2004*), a type 2C protein phosphatase involved in ABA signal transduction, does not appear to dephosphorylate the Ser-30 residue of the AKS1 transcription factor (*Takahashi et al., 2017*). PP2A phosphatases may dephosphorylate AKS1 (*Bu et al., 2017*). The K252a-induced reduction in the SNACS FRET ratio in guard cells provides evidence that the SNACS reporter is reversible. The potential reversibility of SNACS suggests that dynamic up- and down-regulation of SnRK2 protein kinases can be investigated in intact plant cells in real-time.

Methyl jasmonate (MeJA) is a plant hormone that functions in many developmental processes and is important for plant defense responses (*Melotto et al., 2017*). However, MeJA appears to cause different regulation outcomes of stomata depending on conditions. MeJA-mediated stomatal closure was observed in several species including *Paphiopedilum* (*Gehring, 1997*), *Vicia faba* (*Liu et al., 2002*) and *Arabidopsis* (*Suhita et al., 2004*; *Hossain et al., 2011*; *Munemasa et al., 2007*; *Munemasa et al., 2019*; *Yin et al., 2016*; *Förster et al., 2019*). However, other studies reported that MeJA did not promote stomatal closure (*Montillet et al., 2013*; *Savchenko et al., 2014*; *Melotto et al., 2017*; *Zhu et al., 2020*). In addition, the SnRK2.6/OST1 protein kinase is required for MeJA-induced stomatal closure, due to an impaired MeJA-induced stomatal closure response in *snrk2.6/ost1* mutants (*Yin et al., 2016*). Furthermore, *in-gel* kinase assays of guard cell-enriched epidermal tissues suggest that MeJA does not activate SnRK2.6/OST1 activity even though OST1/SnRK2.6 mutation impairs MeJA-induced stomatal closing (*Munemasa et al., 2019*). In the present study, the SNACS reporter did not show clear increases in FRET ratios in guard cells upon 20 μM MeJA application, consistent with this study using guard cell-enriched epidermal tissues (*Munemasa et al., 2019*; *Figure 7*). These findings may support a recently proposed calcium-dependent parallel MeJA response pathway (*Förster et al., 2019*). The SNACS sensor reported here can be used to further investigate MeJA signaling under diverse conditions.

High $CO_2$ and ABA trigger rapid stomatal closure. Gas exchange, stomatal movement and ion channel regulation studies with *ost1* mutant alleles demonstrated that the SnRK2.6/OST1 protein kinase plays a role in $CO_2$-induced stomatal closure (*Xue et al., 2011*; *Merilo et al., 2013*; *Hsu et al., 2018*). Surprisingly however, *in-gel* kinase assays suggested that $CO_2$ elevation did not enhance SnRK2.6/OST1 activity in guard cell protoplasts, even though high $CO_2$ activated S-type anion channels in the same system (*Hsu et al., 2018*). It remained unclear however whether elevated $CO_2$ can activate SnRK2 protein kinase activity in live intact guard cells. In the present study, we used the new SNACS reporter to investigate $CO_2$ signal transduction in intact stomatal guard cells. Time-resolved FRET analyses in guard cells show that $CO_2$ elevation does not cause measurable increases in the FRET ratio of SNACS-expressing guard cells, whereas subsequent exposure to ABA caused clear FRET ratio increases (*Figure 8A and B*), providing evidence in intact guard cells that high $CO_2$ does not activate the SnRK2.6/OST1 activity.

Studies have shown that ABA receptors and OST1/SnRK2.6 are required for wild-type like high $CO_2$-induced stomatal closure (*Chater et al., 2015*; *Xue et al., 2011*; *Hsu et al., 2018*). Classical studies suggested that non-stressed guard cells have a higher ABA concentration than other leaf cells (*Lahr and Raschke, 1988*) and more recently this has been shown experimentally (*Hsu et al., 2018*; *Yoshida et al., 2019*). A basal SnRK2 kinase activity was hypothesized in non-stressed guard cells to explain all experimental observations (*Hsu et al., 2018*). However, direct measurements of the basal SnRK2 activity in guard cells had not been possible. Interestingly, the protein kinase inhibitor K252a caused a clear decrease in FRET ratio of the SNACS reporter and K252a inhibited ABA-induced FRET changes of SNACS in guard cells (*Figure 6A and D*). Furthermore, the ABA-induced SNACS response is disrupted in the *snrk2.2/2.3/2.6* triple mutant (*Figure 5D*). These data provide experimental evidence that SnRK2 protein kinases have a basal kinase activity in guard cells, even in the absence of exogenous ABA application. The basal SnRK2 activity found here could provide an explanation why *ost1/snrk2.6* mutant alleles are clearly impaired in $CO_2$-induced stomatal closing (*Xue et al., 2011*; *Hsu et al., 2018*; *Merilo et al., 2013*), but elevated $CO_2$ does not enhance OST1/ SnRK2.6 activity (*Figure 8*): The present data suggest that basal SnRK2 activity, functioning parallel to the $CO_2$ signal transduction branch, is required for and/or could amplify $CO_2$ signaling. Furthermore, high $CO_2$-induced stomatal closing is slowed but not disrupted in *ost1/snrk2.6* (*Hsu et al., 2018*). This lies in contrast to a stronger ABA-insensitive response of *ost1/snrk2.6* mutant alleles (*Mustilli et al., 2002*; *Yoshida et al., 2002*). These findings are consistent with a parallel role for ABA and OST1 in $CO_2$ signal transduction.

A recent study has suggested that $CO_2$ signal transduction is mediated by the PYL4 and PYL5, but not the PYL2 ABA receptors (*Dittrich et al., 2019*). This conclusion could not be confirmed in the present study using higher order ABA receptor quintuple and sextuple mutant plants, that include gene disruptions of *PYL4* and *PYL5* (*Figure 9*, *Figure 9—figure supplement 1*, *Figure 9— figure supplement 2* and *Figure 9—figure supplement 3*). In the present study, quintuple ABA receptor disruption lines *pyl-11458* and *pyl-12458* were those investigated for gas exchange in *Dittrich et al., 2019*. However, data for the guard cell complementation lines presented here were based on *pyl-11458* whereas gas exchange data in *Dittrich et al., 2019* were based on complementation of *pyl-12458*. Protein expression levels of transgenically-expressed receptors may vary, affecting interpretations. Therefore, use of gene disruption lines in the present study allows for more reliable conclusions and demonstrates that $CO_2$-induced stomatal closing remains robust after *PYL4* and *PYL5* disruption (*Figure 9* and *Figure 9—figure supplement 1*). Furthermore, quantitative differences in protein expression levels of transgenically-expressed ABA receptors in guard cells and growth conditions may affect the outcome of stomatal response data given the relevance of basal ABA concentrations and SnRK2 activity found here. The large transpiration rate in the *pyl-12458* quintuple mutant was reduced by guard cell expression of *PYL5* (Supplementary Figure 8C in *Dittrich et al., 2019*), which may be explained by the response to basal ABA. Consistent with this model, we observed a slowed stomatal opening in *pyl-11458/PYL5* leaves (*Figure 9—figure supplement 2D*). Stomatal responses to high $CO_2$ were affected and slowed in higher order *pyr/pyl/rcar* ABA receptor mutants (*Hsu et al., 2018*). However, data for complementation with single receptors (*Dittrich et al., 2019*) may depend on growth conditions, as the present independent findings in our two laboratories (EM and HK lab and JS lab) and previous studies (*Merilo et al., 2013*; *Hsu et al., 2018*) could not support the model that PYL4 and PYL5 mediate the stomatal $CO_2$ response.

Normalization is often used to display stomatal responses of mutants with largely different steady-state stomatal conductances. Note that, apart from comparing the time dependence of stomatal responses, conclusions derived from interpretation of normalized stomatal conductance data can be equivocal. Mutants that respond to stimulation, but have a higher stomatal conductance and retain a higher stomatal conductance after the response (which may be due to stomatal development and apertures), can show an apparent reduced responsiveness when data are normalized to the values before the onset of a stimulus. As an example, in the present study analyses of normalized stomatal conductance data would lead to inaccurate conclusions: For example, when stomatal conductance data from *Figure 9* were analyzed, but in normalized relative units, that is by dividing the absolute reduction in stomatal conductance with the stomatal conductance at the onset of elevated $CO_2$, the opposite conclusions to non-normalized data analyses could be made (*Figure 9—figure supplement 1*): After normalization the PYR/RCAR sextuple mutant and the *pyl-11458* quintuple mutant had the smallest normalized reductions in stomatal conductance in response to $CO_2$ elevation (*Figure 9—figure supplement 1*). In contrast however, for the same data, the absolute stomatal conductance values show the strongest $CO_2$ responses in the ABA receptor *pyl-12458* quintuple and *pyl-112458* sextuple mutants (*Figure 9A and B*, *Figure 9—figure supplement 1*). These analyses indicate that in order to quantitatively determine whether the stomata of plant lines with different initial stomatal conductances are sensitive to stimuli, it is necessary to apply various kinetic analysis based on non-normalized real-time stomatal conductances. Showing both absolute and normalized data side-by-side (*Hsu et al., 2018*), or providing steady-state stomatal conductance values, can help in developing more robust mechanistic models.

In conclusion, SnRK2s are key protein kinases that mediate abiotic stress responses. Together, the presented research provides a new approach to investigate real-time SnRK2 kinase activity and stress responses in living intact plant cells and tissues. The SNACS reporter could be used to investigate many stress signaling models, and cross talk among pathways (*Yoshida et al., 2006*; *Wang et al., 2018*; *Takahashi et al., 2020*; *Lin et al., 2020*). Furthermore, subcellular targeting of SNACS should be interesting for future comparisons of cellular compartment-localized SnRK2 protein kinase regulation. Furthermore, we provide evidence for the reversibility of SNACS in vivo upon inhibition of kinase activities, suggesting that SnRK2 protein kinase down-regulation can be analyzed as well. Moreover, both elevated $CO_2$ and MeJA did not cause SNACS fluorescence emission ratio increases, providing experimental evidence that these stimuli do not rapidly enhance SnRK2.6/OST1 activity in guard cells as found in real-time analyses here. Furthermore, our gas exchange analyses in intact plants and leaves, could not confirm the recently proposed model that stomatal $CO_2$ signaling requires the PYL4 and PYL5 ABA receptors (*Dittrich et al., 2019*). The present findings provide in vivo evidence for $CO_2$ signal transduction mechanisms in stomatal signaling that use basal ABA signaling and basal OST1 activity, but do not rapidly upregulate OST1/SnRK2 kinase activity.

## Materials and methods

### Key resources table

| Reagent type (species) or resource | Designation | Source or reference | Identifiers | Additional information |
|---|---|---|---|---|
| Gene (*Arabidopsis*) | AKS1 | Tair (https://www.arabidopsis.org/) | Tair ID: At1g51140 | |
| Gene (*Arabidopsis*) | 14-3-3, GF14phi | Tair (https://www.arabidopsis.org/) | Tair ID: At1g35160 | |
| Strain, strain background (*Escherichia coli*) | BL21-CodonPlus (DE3) | Agilent Technologies | Model: 230245 | Electro-competent cells |
| Strain, strain background (*Agrobacterium tumefaciens*) | GV3101 | Other | | Widely distributed |
| Antibody | Anti-FLAG M2 Mouse monoclonal | Sigma-Aldrich | RRID:AB_262044 | x5,000 |
| Chemical compound, drug | K-252a | Sigma-Aldrich | Cas No. 99533-80-9 | |

*Continued on next page*

*Continued*

| Reagent type (species) or resource | Designation | Source or reference | Identifiers | Additional information |
|---|---|---|---|---|
| Chemical compound, drug | W-7 | Sigma-Aldrich | Cas No. 61714-27-0 | |
| Chemical compound, drug | MeJA | Bedoukian Research, Inc | Ct. 06810–4192 | |
| Software, algorithm | MetaFluor software | MetaFluor (https://www.moleculardevices.com/products/cellular-imaging-systems/acquisition-and-analysis-software/metamorph-microscopy) | RRID:SCR_014294 | version 7.0r3 |
| Software, algorithm | Fiji software | Fiji (https://imagej.net/Fiji) | RRID:SCR_002285 | |
| Software, algorithm | GraphPad Prism software | GraphPad Prism (https://graphpad.com) | RRID:SCR_015807 | version 7.0 |

## Construction of SNACS reporter

The plasmid backbone used to construct SNACS reporter plasmids was the SOMA construct as reported by *Zaman et al., 2019*. Restriction enzyme cloning was used: the FHA domain in the original vector was replaced with the full-length 267 amino acid coding region of the 14-3-3 protein, GF14phi (At1g35160) and the substrate domain of the original plasmid was replaced with DNA encoding for amino acids 1–48 of the *Arabidopsis* AKS1 protein (At1g51140) (*Takahashi et al., 2017*). To generate the phospho-site mutant isoform of the SNACS reporter, site-directed mutagenesis was performed to change the coding sequence at the AKS1 serine-30 to alanine resulting in the SNACS$^{S785A}$ reporter. In addition, the StrepII-tag was inserted into the C-terminus via the XbaI site for expression of recombinant SNACS protein in *E. coli*. For expression in plants, the SNACS coding fragment was PCR-amplified with the attB1 and attB2 adaptor primers. This fragment was introduced into the donor vector pDONR221 using BP clonase (Invitrogen). Final destination vectors for expression in plants were obtained by using a multisite gateway recombination system, using the pH7m34GW destination vector and p35S/pDONRP4-P1R constructs. Vectors carrying *35S: SNACS* or *35S: SNACS$^{S785A}$* were used for plant transformation.

## In vitro FRET analyses and phosphorylation assay

The recombinant SNACS protein, SNACS$^{S785A}$ protein, GST-OST1/SnRK2.6, GST-OST1$^{D140A}$, GST-CPK6 and GST-SnRK2.3 were expressed in BL21-CodonPlus (DE3)-RIL cells (Stratagene). StrepII-tagged proteins were purified using Strep-Tactin Macroprep columns (IBA). GST-tagged recombinant proteins were purified using Glutathione Sepharose 4B. 9 µg of the SNACS or SNACS$^{S785A}$ were added to a reaction containing 1 × PKA buffer (50 mM Tris-HCl pH7.5, 10 mM MgCl$_2$), 0.2 mM ATP-Na, 2 mM DTT and water to a total volume of 50 µl with 2 µM free Ca$^{2+}$ buffered only for GST-CPK6. For reactions including GST-OST1, GST-OST1$^{D140A}$, GST-CPK6, and GST-SnRK2.3, 4 µg of these proteins were also added. The reactions were then incubated at room temperature for 2 hr and the fluorescence emission spectrum was measured using a TECAN SPARK multimode plate reader. Excitation was performed at 434 nm and the emission range analyzed was 460 to 560 nm.

In vitro phosphorylation assays were performed as previously described (*Takahashi et al., 2017*). Recombinant SNACS reporter proteins were incubated in phosphorylation buffer (50 mM Tris-HCl, 10 mM MgCl$_2$, 0.1% TritonX-100, 1 mM DTT, pH7.5) with recombinant GST-tagged OST1/SnRK2.6 or CPK6 for 30 min at room temperature in the presence of 1 µCi [γ-$^{32}$P]-ATP and 200 µM ATP. The reactions were stopped by the addition of SDS-loading buffer. After separation with 10% SDS-polyacrylamide gels, proteins were visualized by coomassie blue staining, and phosphorylated proteins were visualized by autoradiography.

## Infiltration of *Nicotiana benthamiana* for transient expression

SNACS driven by the CaMV 35S promoter was co-expressed with *pUBQ10:OST1-HF* via co-infiltration using the GV3101 strain of *Agrobacterium tumefaciens*. In parallel, we co-infiltrated *N. benthamiana* leaves using Agrobacterium carrying the p19 suppressor of gene silencing to enhance

transgene expression. *N. benthamiana* leaves from 3-week-old plants were used for infiltration. After 3 days of infiltration, microscope imaging analyses were performed (see below).

## Transgenic *Arabidopsis* lines and plant growth

SNACS and SNACS[S785A] reporters carrying plasmids were transformed into the *Arabidopsis* Columbia 0 accession expressing, *pUBQ10:OST1-HF* in the *ost1-3* genetic background (*Waadt et al., 2015*), *ost1-3* (salk_008068) (*Yoshida et al., 2002*), *snrk2.2/2.3* (GABI-Kat_807G04/salk_107315) and *snrk2.2/3/6* (GABI-Kat_807G04/salk_107315/salk_008068) (*Brandt et al., 2015*) via the floral dip method using the GV3101 strain of *Agrobacterium tumefaciens* (*Zhang et al., 2006*). Primary transformants expressing the sensor constructs were selected on 0.5 MS supplemented with 25 µg/mL hygromycin and further cultivated in soil in a growth room (16 h day/8 hr night). We selected positive transformants by fluorescence intensity using confocal microscopy as described below. Transgenic *Arabidopsis* lines used in the present study are listed in *Supplementary file 1*.

## Sample preparation and imaging analyses

For *N. benthamiana* and *Arabidopsis* imaging, detached *N. benthamiana* and *Arabidopsis* leaves were prepared with the abaxial leaf epidermises on a cover glass using medical adhesive (Holliser, Libertyville, IL). A razor blade was then used to carefully remove the upper mesophyll cell layers of *N. benthamiana* and *Arabidopsis* leaves to yield intact epidermal strips (*Young et al., 2006*), which were further incubated in 2 mL assay buffer (5 mM KCl, 50 µM $CaCl_2$, 10 mM MES-Tris, pH 5.6) for an additional 1 hr. 4- to 6- week-old transgenic *Arabidopsis* plants were used. FRET ratio-imaging was conducted as previously described (*Waadt et al., 2014*; *Allen et al., 1999*), with the difference of using 150 ms exposures to reduce bleaching of the fluorescent proteins. Ratiometric measurements were conducted by interchanging the following band-pass emission filters 480 nm (DF30) and 535 nm (DF25) with a computer-controlled filter wheel (*Allen et al., 1999*). Excitation light was 434 nm (DF20) (*Allen et al., 1999*). Images were acquired in intervals of 6 s using MetaFluor software. Image analyses and processing were conducted using Fiji with the following applications: Background subtraction, gaussian blur, 32-bit conversion, threshold, ratio calculation and physics look up (*Schindelin et al., 2012*). Whole individual stomata including the central stomatal pore were selected as regions of interest (ROI) for individual stomate ratio imaging using Fiji and the average ratio value within that ROI was then measured. For background subtraction, epidermal pavement cell background regions were analyzed in which no evident fluorescence of the SNACS reporter was evident. The fluorescence intensity of the SNACS reporter in guard cells appeared to be higher than that in epidermal pavement cells or the SNACS expression levels in the cytoplasm of guard cells appeared to be higher than in surrounding cells. Note that the focal plane during imaging in the present study was focused on guard cells, which is slightly shifted from the focal plane of epidermal pavement cells (*Figure 2—figure supplement 1*). Additionally, since we investigated SNACS-dependent FRET responses in stomata in the present study, we choose microscopic regions in some experiments showing SNACS fluorescence in guard cells with apparent low SNACS fluorescence in surrounding epidermal cells. Therefore, at the illustrated image gains, epidermal cells are not clearly visible. The FRET ratio of YPet to Turquoise GL emission fluorescence was normalized to the average over first 5 min before the indicated treatments. For the indicated treatment applications, epidermal strips were perfused by gently pipetting 5 to 6 times with the assay buffer supplemented with the indicated final concentrations of treatment (ABA, K252a, W-7, MeJA and solvent controls). $CO_2$ responses of the SNACS reporter were performed as previously described (*Hsu et al., 2018*) except that solutions for high $CO_2$ were bubbled with air containing 2000 ppm $CO_2$. The final $CO_2$ concentrations in the recording chamber of 115 ppm and 1170 ppm after bath perfusion from continuously bubbled solutions to the imaging chamber via tubing and a peristaltic pump were determined as described previously (*Young et al., 2006*). For ABAleon2.15 reporter imaging, experiments were performed as described above, except that images were acquired in intervals of 12 s. The FRET ratio of YPet to Turquoise GL emission fluorescence was normalized to the average over the first 10 min for $CO_2$ response experiments. Paired t-test analyses were performed using Graphpad Prism version 7.0.

## Whole plant gas exchange experiments

25–30 days old *Arabidopsis* plants grown in pots containing 2:1 (v:v) peat: vermiculite mixture kept in Snijders chambers (Snijders Scientific, Drogenbos, Belgia) at 12/12 photoperiod, 23/18 ˚C temperature, 160 µmol m$^{-2}$ s$^{-1}$ light and 70% relative humidity, were used for gas exchange experiments. Custom-built 8-chamber temperature-controlled gas-exchange device as described before (*Kollist et al., 2007*) was used to measure water vapor concentrations in the air entering and leaving the measurement chambers and to calculate the values of whole-plant stomatal conductance. Standard conditions during the 1–2 hr stabilization period were: ambient $CO_2$ (~400 ppm), light 160 µmol m$^{-2}$ s$^{-1}$, relative air humidity (RH) ~61 ± 3%. Then, the $CO_2$ concentration was increased to 800 ppm for 50 min. Plant area was calculated from photographs using ImageJ 1.37 v (National Institutes of Health, USA). Stomatal conductance to water vapor is calculated as described in *von Caemmerer and Farquhar, 1981*, with details of the calculation procedure having been described earlier (*Kollist et al., 2007*). In order to compare the $CO_2$ responses of different lines, we calculated closure half-times and rapid high $CO_2$-induced changes in stomatal conductance. The whole 50 min stomatal response was scaled to a range of 0–100% and the time when 50% of stomatal closure was reached was calculated for half-response times. Rapid high $CO_2$-induced changes in stomatal conductance were calculated as the differences in the values of stomatal conductance 20 min after elevated $CO_2$ was applied and pretreatment stomatal conductance values.

## Time-Resolved intact leaf stomatal conductance experiments

ABA receptor quintuple mutant, *pyr1/pyl1/pyl4/pyl5/pyl8* (*Antoni et al., 2013*), and guard cell-targeted ABA receptor complemented plants, *pyr1/pyl1/pyl4/pyl5/pyl8/pGC1::PYL1* (*pyl-11458/PYL1*), *pyr1/pyl1/pyl4/pyl5/pyl8/pGC1::PYL4* (*pyl-11458/PYL4*), and *pyr1/pyl1/pyl4/pyl5/pyl8/pGC1::PYL5* (*pyl-11458/PYL5*) were used for gas exchange analyses (*Antoni et al., 2013*; *Dittrich et al., 2019*). Stomatal conductance recordings from intact leaves of 5- to 6.5-week-old plants were conducted starting 1 to 2 hr after growth chamber light onset. A Li-6400XT infrared (IRGA)-based gas exchange analyzer system was used with an integrated 6400-02B LED Light Source (Li-Cor Inc). The measurement protocols were similar to those described by *Dittrich et al., 2019*. Leaves were clamped and kept in the dark at 400 ppm ambient $CO_2$ (*Dittrich et al., 2019*), 21˚C, 65–70% relative air humidity, and 400 µmol s$^{-1}$ flow rate until the stomatal conductance stabilized. For stomatal responses to light and [$CO_2$] shifts, stomatal conductance was first measured at 400 ppm ambient $CO_2$ in the dark for 10 mins; then exposed to continuous light intensity 125 µmol m$^{-2}$ s$^{-1}$ (*Dittrich et al., 2019*). Approximately 80 mins after light-on, ambient [$CO_2$] was increased to 1000 ppm (*Dittrich et al., 2019*) to analyze stomatal responses to high [$CO_2$]. Following gas exchange experiments, the area of each analyzed leaf was measured for stomatal conductance calculations.

Higher order ABA receptor mutants in the present study were genotyped (*Figure 9—figure supplement 3*). Because the guard cell-targeted ABA receptor complementation lines were segregating (*Dittrich et al., 2019*), the *pGC1::PYL1*, *pGC1::PYL4*, and *pGC1::PYL5* transgene in each plant analyzed by gas exchange was subsequently confirmed by PCR in the present study (*Figure 9—figure supplement 3*). We have also isolated homozygous *pyr1/pyl1/pyl4/pyl5/pyl8/pGC1::PYL1*, *pyr1/pyl1/pyl4/pyl5/pyl8/pGC1::PYL4*, and *pyr1/pyl1/pyl4/pyl5/pyl8/pGC1::PYL5* lines. The seeds of the homozygous higher order ABA receptor mutants and homozygous guard cell-targeted ABA receptor complementation lines used in this study will be donated to the Arabidopsis Biological Resource Center.

## Acknowledgements

We thank Dr. Alexandra Newton for discussions, Dr. Rainer Waadt for general advice on FRET imaging and Dr. Pedro L Rodriguez for providing the original ABA receptor mutant seeds. This research was funded by grants from the National Institutes of Health (GM060396) and the National Science Foundation to JIS (MCB-1900567). LZ was supported by a China Scholarship Council fellowship, and YT was supported by a Postdoctoral Fellowship for Research Abroad from the Japan Society for the Promotion of Science. Data of EM and HK in *Figure 9* and *Figure 9—figure supplements 1* and *2* were supported by the Estonian Research Council (grants PUT1133, PRG719 and PRG433) and

European Regional Development Fund (Center of Excellence in Molecular Cell Engineering CEMCE). PK was supported by the National Science Foundation (NSF MCB-1137950).

## Additional information

### Funding

| Funder | Grant reference number | Author |
|---|---|---|
| National Science Foundation | MCB-1900567 | Julian I Schroeder |
| National Institutes of Health | GM060396 | Julian I Schroeder |
| China Scholarship Council | | Li Zhang |
| Japan Society for the Promotion of Science | | Yohei Takahashi |
| Eesti Teadusagentuur | PUT1133 | Ebe Merilo |
| Eesti Teadusagentuur | PRG719 | Ebe Merilo |
| Eesti Teadusagentuur | PRG433 | Hannes Kollist |
| European Regional Development Fund | Center of Excellence in Molecular Cell Engineering (CEMCE) | Hannes Kollist |
| National Science Foundation | MCB-1137950 | Patrick J Krysan |

The funders had no role in study design, data collection and interpretation, or the decision to submit the work for publication.

### Author contributions

Li Zhang, Investigation, Writing - original draft; Yohei Takahashi, Conceptualization, Investigation, Methodology, Writing - original draft, Writing - review and editing; Po-Kai Hsu, Hannes Kollist, Ebe Merilo, Investigation, Methodology, Writing - original draft, Writing - review and editing; Patrick J Krysan, Methodology, Writing - review and editing; Julian I Schroeder, Supervision, Funding acquisition, Methodology, Writing - original draft, Project administration, Writing - review and editing

### Author ORCIDs

Li Zhang (iD) https://orcid.org/0000-0003-0467-0290
Yohei Takahashi (iD) https://orcid.org/0000-0002-9406-4093
Po-Kai Hsu (iD) https://orcid.org/0000-0001-7265-7077
Julian I Schroeder (iD) https://orcid.org/0000-0002-3283-5972

### Decision letter and Author response
Decision letter https://doi.org/10.7554/eLife.56351.sa1
Author response https://doi.org/10.7554/eLife.56351.sa2

## Additional files

### Supplementary files
• Supplementary file 1. Transgenic lines used in this study. Detailed information on the transgenic lines is provided including the plasmid, promoter, and genetic background.

• Supplementary file 2. Primer sequences for genotyping. Primers used to genotype higher order ABA receptor mutants (*Figure 9—figure supplement 3*).

• Transparent reporting form

### Data availability
Data generated or analysed during this study are included in the manuscript and supporting files.

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
