## [Decision Letter]

**Acceptance summary:**

Plant cells must integrate a number of external signals to make effective cellular decisions. Tracking the precise molecular pathway by which they do this can be challenging as genetic approaches are often indirect. In this study, a new sensor for activation of the SnRK2/OST1 protein kinase in living *Arabidopsis* cells was generated, and provides a way to better assess direct and indirect signaling inputs into cellular responses.

**Decision letter after peer review:**

Thank you for submitting your article "FRET biosensor development shows SnRK2 kinase activation by ABA but not by MeJA and elevated CO_2_ during stomatal closure" for consideration by *eLife*. Your article has been reviewed by three peer reviewers, one of whom is a member of our Board of Reviewing Editors, and the evaluation has been overseen by Christian Hardtke as the Senior Editor. The following individual involved in review of your submission has agreed to reveal their identity: Jeffrey Leung (Reviewer #3).

The reviewers have discussed the reviews with one another and the Reviewing Editor has drafted this decision to help you prepare a revised submission.

As the editors have judged that your manuscript is of interest, but as described below that additional experiments are required before it is published, we would like to draw your attention to changes in our revision policy that we have made in response to COVID-19 (https://elifesciences.org/articles/57162). First, because many researchers have temporarily lost access to the labs, we will give authors as much time as they need to submit revised manuscripts. We are also offering, if you choose, to post the manuscript to bioRxiv (if it is not already there) along with this decision letter and a formal designation that the manuscript is 'in revision at *eLife*'. Please let us know if you would like to pursue this option. It was the conclusion of the reviewers and BRE that the types of experiments that needed to be added were controls that would have been expected for the complete characterization of a new sensor and so the authors would have been expected to have done these (and may simply not included them in pursuit of brevity).

Summary:

This manuscript reports in vivo analysis of SnRK2 activation of *Arabidopsis* in response to diverse stimuli, enabled by the creation of a new FRET-based reporter for SnRK2 kinase activity (SNACS). The SnRK2.6/OST1 is a key regulator in ABA-induced stomatal closure. OST1 deficiency also impairs CO_2_- and meJA-triggered stomatal responses. Whether CO_2_- and meJA use the ABA signal pathway upstream of SnRK2s, (i.e. ABA receptors and the associated PP2Cs) is highly debated. The authors suggest that, based on SNACS, there is normally some basal activity of SnRK2 in Arabidopsis guard cells, and that this activity is enhanced upon application of ABA, but not of MeJA and high [CO_2_] suggesting that the stomatal opening and closing responses associated with these chemicals can be uncoupled from SnRK2 activation. Upon finding that high [CO_2_] failed to affect SNACS, they also revisited a recent high-profile result that suggested the role of ABA receptor proteins (PYR/PYLs) in stomatal responses to high [CO_2_]. Here, the analysis of multiple ABA receptor-deficient Arabidopsis mutants showed proper stomatal closing responses towards doubled [CO_2_], contradicting that previous paper. Based in the gas exchange measurements reported here and the previous track record of these authors, the reviewers placed confidence in this new result.

The results characterizing and in vivo SnRK2 activity sensor are exciting and a major advance in the field. SNACS could be very useful for in vivo testing of responses to environmental stimuli, and bridging the gap between in vivo observations that SnRK2 mutants are insensitive to stimuli with in vitro assays that failed to show that these stimuli change SnRK2 kinase activity.

While there was excitement about the opportunities the SNACS sensor provides, there were also a number of experimental parameters, controls or tests that would be required for confidence in this tool. These are enumerated below. In addition, a number of smaller modifications are needed for clarity.

Essential revisions:

Confirmation of behaviors of the SNACS reporter. One "selling point" of SNACS is that it could report responses to many stimuli in many cells. Yet, the expression pattern and other attributes of SNACs are not described sufficiently.

1) Detail the expression pattern and subcellular localization of SNACs and discuss caveats/concerns related to the expression. For example, it appears that the 35S promoter was used for expression of SNACS, yet we are only shown zoomed in images of GCs. Where else is SNACS expressed and was response to stimuli tested in any other tissue?

2) Discuss the limitations or alternatives to a reporter that uses this one particular substrate, and especially one that would normally be in the nucleus? Is there precedent for reporters behaving differently in cell compartments (nucleus vs. cytoplasm vs. PM-adjacent) that need to be discussed here? Are there other known SnRK substrates/14-3-3 pairs? In Figure 3C-E it is not clear whether the reporter is nuclear or not, whereas in Figure 2, the signal looks cytoplasmic. Because PP2Cs and PYLs are PM associated and may directly regulate associated SnRK2s and ion channel targets, would such a signal cascade at the membrane be missed because it did not lead to a major change of an abundant, soluble SNAC reporter?

3) What is the sensitivity of the SNAC to monitor 'SnRK2 activation' in vivo given that SNAC records the phosphorylation status of a single SnRK2 target, and phosphorylation status is of course controlled by phosphorylation and dephosphorylation. ABA may not only activate OST1 but could also deactivate the unknown protein phosphatase(s) of the SNAC target. To partly address the issue, the dose response of ABA, stomatal aperture, and the FRET SNAC signal would be required. Also, in this context, it was not clear to reviewers what the response relationship is between the level of SNAC phosphorylation and FRET signal ratio in the in vitro analysis (Figure 1).

4) Make it clear how the FRET signals are being reported. It is confusing why we only see GCs in the images of Figures 3 and 4. Is there no activity in surrounding cells? Because in Figure 2, cells surrounding the transformed cells are blue and I assumed that was because these cells were not expressing the reporter, but if the color scheme is relative wouldn't they be green (no change) like in Figures 3 and 4?

---

## [Author Response]

Summary:This manuscript reports in vivo analysis of SnRK2 activation of Arabidopsis in response to diverse stimuli, enabled by the creation of a new FRET-based reporter for SnRK2 kinase activity (SNACS). The SnRK2.6/OST1 is a key regulator in ABA-induced stomatal closure. OST1 deficiency also impairs CO_2_- and meJA-triggered stomatal responses. Whether CO_2_- and meJA use the ABA signal pathway upstream of SnRK2s, (i.e. ABA receptors and the associated PP2Cs) is highly debated. The authors suggest that, based on SNACS, there is normally some basal activity of SnRK2 in Arabidopsis guard cells, and that this activity is enhanced upon application of ABA, but not of MeJA and high [CO_2_] suggesting that the stomatal opening and closing responses associated with these chemicals can be uncoupled from SnRK2 activation. Upon finding that high [CO_2_] failed to affect SNACS, they also revisited a recent high-profile result that suggested the role of ABA receptor proteins (PYR/PYLs) in stomatal responses to high [CO_2_]. Here, the analysis of multiple ABA receptor-deficient Arabidopsis mutants showed proper stomatal closing responses towards doubled [CO_2_], contradicting that previous paper. Based in the gas exchange measurements reported here and the previous track record of these authors, the reviewers placed confidence in this new result.

We agree with the conclusion that part of our findings address this question and could not support a main conclusion of a recent report of Dittrich et al., 2019. We have added brief text to clarify this point to the Abstract, Introduction and Results. We have also made small edits to this section for clarity in the Discussion (eighth paragraph), which already had discussed the difference to the recent report.

Essential revisions:Confirmation of behaviors of the SNACS reporter. One "selling point" of SNACS is that it could report responses to many stimuli in many cells. Yet, the expression pattern and other attributes of SNACs are not described sufficiently.1) Detail the expression pattern and subcellular localization of SNACs and discuss caveats/concerns related to the expression. For example, it appears that the 35S promoter was used for expression of SNACS, yet we are only shown zoomed in images of GCs. Where else is SNACS expressed and was response to stimuli tested in any other tissue?

In reference to SNACS subcellular localization, we see clear fluorescence in the cytoplasm, including in cytoplasmic spaces or strands between vacuoles. We observe a weak or no fluorescence in the nucleus. As we have not yet targeted SNACS to the nucleus specifically, we have added text in our manuscript to discuss this comment and now point out that targeting SNACS to subcellular compartments could be done in the future (subsection “SNACS reports SnRK2 activity dynamics in plant cells**”;** Discussion, second and last paragraphs; subsection “Sample preparation and imaging analyses”). We have also added a new figure (Figure 3—figure supplement 1) showing fluorescence from the YPet channel. In reference to the SNACS expression pattern, we added this sentence: "SNACS fluorescence was observed throughout plant seedlings in the *35S:SNACS* expression lines, including guard cells and leaf epidermal cells (Figure 2—figure supplement 1)." We have also added a figure to illustrate epidermal cell expression (Figure 2—figure supplement 1). We have added these data from existing data, due to the COVID-19 lock down. We also discuss that guard cell fluorescence appears to be stronger than in leaf pavement cells (subsection “Sample preparation and imaging analyses”).

2) Discuss the limitations or alternatives to a reporter that uses this one particular substrate, and especially one that would normally be in the nucleus? Is there precedent for reporters behaving differently in cell compartments (nucleus vs. cytoplasm vs. PM-adjacent) that need to be discussed here? Are there other known SnRK substrates/14-3-3 pairs?

For our SNACS construct, we have excluded the AKS1 bHLH domain which is important for nuclear localization of AKS1 and its function as a transcription factor. We have added this information to the Discussion: "The full length AKS1 protein functions as a transcription factor in the nucleus. However, the bHLH domain of AKS1, which is important for its transcription factor function and nuclear localization, was excluded from our SNACS sensor construct. SNACS fluorescence appears to be observed mainly in the cytoplasm and in cytoplasmic spaces between vacuoles, with possible weak or no expression in the nucleus (Figure 3—figure supplement 1)."

We agree that targeting SNACS to cellular compartments would be an interesting approach. We have added a sentence to the Discussion: "Furthermore, subcellular targeting of SNACS should be interesting for future comparisons of cellular compartment-localized SnRK2 protein kinase regulation."

In reference to the SnRK2 substrate and 14-3-3 protein binding, presently the AKS family is the only or one of few published example that has been shown to exhibit a phosphorylation-induced interaction in plant cells in response to ABA to our knowledge. The proximity and linking of both protein domains in the SNACS construct likely enhances the probability that these partners would interact relative to other proteins in the investigated cells.

In Figure 3C-E it is not clear whether the reporter is nuclear or not, whereas in Figure 2, the signal looks cytoplasmic.

We agree with the reviewers. It appears that SNACS in general is more strongly expressed in the cytoplasm. In some guard cells it appears that nuclear expression may occur but may be weak or fluorescence may be “bleeding” through from surrounding regions. To make this point more clear, we have added an image showing guard cell YPet fluorescence. We have added text and a new figure (Figure 3—figure supplement 1) referring to this observation and as described above now discuss the need for future research into compartment-localized SnRK2 activation analyses.

Because PP2Cs and PYLs are PM associated and may directly regulate associated SnRK2s and ion channel targets, would such a signal cascade at the membrane be missed because it did not lead to a major change of an abundant, soluble SNAC reporter?

It is unclear whether SNACS associates with the PM or not. In the future modified SNACS sensors could be used to visualize subcellular localized ABA signaling, including a PM-localized SNACS. We added a sentence "Furthermore, subcellular targeting of SNACS should be interesting for future comparisons of cellular compartment-localized SnRK2 protein kinase regulation." in the Discussion section. We note that ABA clearly causes FRET shifts of SNACs in our many experiments and the mechanisms upstream in CO_2_ signaling are not exclusively at the plasma membrane, including CA1, MPK4/12, and CBC1/CBC2.

3) What is the sensitivity of the SNAC to monitor 'SnRK2 activation' in vivo given that SNAC records the phosphorylation status of a single SnRK2 target, and phosphorylation status is of course controlled by phosphorylation and dephosphorylation. ABA may not only activate OST1 but could also deactivate the unknown protein phosphatase(s) of the SNAC target. To partly address the issue, the dose response of ABA, stomatal aperture, and the FRET SNAC signal would be required. Also, in this context, it was not clear to reviewers what the response relationship is between the level of SNAC phosphorylation and FRET signal ratio in the in vitro analysis (Figure 1).

In the revised manuscript, we have added additional data showing that adding two times the concentration of the OST1 protein kinase barely affected the FRET ratio shift (Figure 1—figure supplement 1E). We have not conducted the proposed experiments in planta, and also it is uncertain whether we could try them soon because of the COVID-19 lab lock down. However, in our original manuscript, we showed that in *ost1/snrk2.6* single mutants and in *snrk2.2/snrk2.3* double mutants, ABA caused slow (*ost1-3*) but measurable increases in the FRET ratio of SNACS, but not in *snrk2.2/2.3/2.6* triple mutant guard cells (Figure 5). Furthermore, SNACS-dependent FRET signals were reversed by a protein kinase inhibitor (Figure 6). Together these data suggest that SNACS is sensitive to low baseline SnRK2 activity, and that SNACS can dynamically report biological increases in activity to the physiological stimulus, ABA. To make this point more clear, we have added an explanation in the Discussion (fourth paragraph).

4) Make it clear how the FRET signals are being reported. It is confusing why we only see GCs in the images of Figures 3 and 4. Is there no activity in surrounding cells? Because in Figure 2, cells surrounding the transformed cells are blue and I assumed that was because these cells were not expressing the reporter, but if the color scheme is relative wouldn't they be green (no change) like in Figures 3 and 4?

We updated the Materials and methods section to address this relevant point more clearly: “The fluorescence intensity of the SNACS reporter in guard cells appeared to be higher than that in epidermal pavement cells or the SNACS expression levels in the cytoplasm of guard cells appeared to be higher than in surrounding cells. Note that the focal plane during imaging in the present study was focused on guard cells, which is slightly shifted from the focal plane of epidermal pavement cells (Figure 2—figure supplement 1). Additionally, since we investigated SNACS-dependent FRET responses in stomata in the present study, we choose microscopic regions in some experiments showing SNACS fluorescence in guard cells with apparent low SNACS fluorescence in surrounding epidermal cells. Therefore, at the illustrated image gains, epidermal cells are not clearly visible.” In addition, we have added a figure panel showing SNACS fluorescence in *Arabidopsis* leaf epidermal cells to address this point (Figure 2—figure supplement 1).